# Representation Interference Suppression via Non-linear Value Factorization for Indecomposable Markov Games

## Abstract

Value factorization is an efficient approach for centralized training with decentralized execution in cooperative multi-agent reinforcement learning tasks. As the simplest implementation of value factorization, Linear Value Factorization (LVF) attracts wide attention. In this paper, firstly, we investigate the applicable conditions of LVF, which is important but usually neglected by previous works. We prove that due to the representation limitation, LVF is only perfectly applicable to an extremely narrow class of tasks, which we define as the decomposable Markov game. Secondly, to handle the indecomposable Markov game where the LVF is inapplicable, we turn to value factorization with complete representation capability (CRC) and explore the general form of the value factorization function that satisfies both Independent Global Max (IGM) and CRC conditions. A common problem of these value factorization functions is the representation interference among true Q values with shared local Q value functions. As a result, the policy could be trapped in local optimums due to the representation interference on the optimal true Q values. Thirdly, to address the problem, we propose a novel value factorization method, namely Q Factorization with Representation Interference Suppression (QFRIS). QFRIS adaptively reduces the gradients of the local Q value functions contributed by the non-optimal true Q values. Our method is evaluated on various benchmarks. Experimental results demonstrate the good convergence of QFIRS.

## 1 Introduction

Centralized training with decentralized execution (CTDE) (Lowe et al., 2017; Oliehoek et al., 2008; Foerster et al., 2016) shows surprising performance and great scalability in challenging fully cooperative multi-agent reinforcement learning (MARL) tasks (Tan, 1993b). Such tasks only provide rewards shared by all agents. Each agent is expected to deduce its own contribution to the team, which introduces the problem of credit assignment (Foerster et al., 2018). As a simple and efficient approach for credit assignment in the CTDE paradigm, value factorization, especially Linear Value Factorization (LVF) recently gains growing attention, e.g., VDN (Sunehag et al., 2017) and QMIX (Rashid et al., 2018). An important property of LVF is that it concisely meets the Independent Global Max (IGM) principle (Son et al., 2019). The IGM principle is defined as the identity between the joint Q value function and the set of factorized local Q value functions, which is wildly acknowledged as a critical rule for value factorization.

However, the linearly factorizable joint Q value function in LVF is incapable to represent non-linear true Q value functions, known as the representation limitation of LVF. Recent works focus on the solutions to the representation limitation but usually neglect under what conditions the true Q value function is not linearly factorizable. In this paper, we prove that in the context of Markov games, the linear factorizability relies on two conditions: (1) the reward function is linearly factorizable on a set of subspaces of the joint state-action space; (2) the state transitions in each subspace is irrelevant to the state and action out of the subspace. Based on the two conditions above, we define the decomposability of the Markov game. In words, the true Q value function is linearly factorizable if and only if the Markov game is decomposable. Most of the tasks are indecomposable Markov games, so we go deeper into the property of LVF in this case. We prove that the target of the joint Q

value function in Bellman equation (Sutton & Barto, 2018) is always unbiased for LVF under value iteration in sarsa manner.

To deal with the indecomposable Markov game where the true Q value function is not linearly factorizable, we consider improving the representation capability of the value factorization function by introducing extra approximators. According to the partial derivative on local Q value functions, value factorization functions can be classified into two categories, i.e., linear and non-linear. For both categories, we investigate the conditions of value factorization functions that satisfy the complete representation capability (i.e., the capability to approximate any true Q value function) and the IGM principle. Then we propose a rule to generate qualified functions and list some example functions for both linear and non-linear cases.

A common problem of these value factorization functions is the representation interference among true Q values. Specifically, a local Q value corresponds to multiple true Q values in value factorization. As a result, the representation of these true Q values is interfered by each other through the training of the shared local Q value function. The representation interference on the optimal true Q value function could leave the policy trapped in the local optimum. To address the problem, we design a novel value factorization function. Our method, namely Q Factorization with Representation Interference Suppression (QFRIS), alleviates the representation interference on the optimal true Q value by reducing the weight contributed by the non-optimal ones. QFRIS is evaluated on matrix game, predator-prey and starcraft multi-agent challenge. The experimental results demonstrate the good convergence of our method.

We have three main contributions in this work: (1) We prove a sufficient and necessary condition for the linear factorizability of the true Q value function, which can be used to distinguish whether the joint Q value function of LVF is faced with representation limitation; (2) To deal with indecomposable Markov games, we propose the rules to generate value factorization functions that satisfy both IGM and CRC conditions; (3) We point out a common problem of value decomposition, namely representation interference, and design a novel value factorization function to address the problem. Our method shows good convergence in experiments on various benchmarks.

## 2 PRELIMINARIES

### 2.1 DEC-POMDP

A fully cooperative multi-agent reinforcement learning problem can be modelled by the Decentralized Partially Observable Markov Decision Process (Dec-POMDP), which is usually described by a tuple $\mathcal{G} =< \boldsymbol{\mathcal{S}}, \boldsymbol{\mathcal{U}}, \mathcal{P}, \boldsymbol{r}, Z, O, n, \gamma >$ (Guestrin et al., 2001; Oliehoek & Amato, 2016; Seuken & Zilberstein, 2008). $\boldsymbol{s} \in \boldsymbol{\mathcal{S}}$ denotes the global state of the environment, by which a local observation $z_i \in Z_i$ is assigned to agent $i \in I \equiv \{1, 2, \cdots, n\}$ according to the observation function $O : \boldsymbol{\mathcal{S}} \times I \to Z_i$. After receiving $z_i$, each agent chooses an individual action $u_i \in \mathcal{U}_i$ based on its local policy $\pi_i(u_i|\tau_i) : \mathcal{T}_i \times \mathcal{U}_i \to [0, 1]$, where $\tau_i \in \mathcal{T}_i \equiv (Z_i \times \mathcal{U}_i)$ is the local observation-action history, i.e., the local trajectory. After the execution of the joint action $\boldsymbol{u} = \{u_1, \cdots, u_n\}$, a reward $\boldsymbol{r}$ shared by all agents and the next state $\boldsymbol{s}'$ are generated by the reward function $\boldsymbol{r}(\boldsymbol{s}, \boldsymbol{u}) : \boldsymbol{\mathcal{S}} \times \boldsymbol{\mathcal{U}} \to \mathcal{R}$ and transition function $\mathcal{P}(\boldsymbol{s}'|\boldsymbol{s}, \boldsymbol{u}) : \boldsymbol{\mathcal{S}} \times \boldsymbol{\mathcal{U}} \to \boldsymbol{\mathcal{S}}$, respectively. $\gamma \in [0, 1)$ is a discount factor. Note that we use bold symbols to denote the global and joint variables, e.g., $\boldsymbol{\mathcal{S}}$ and $\boldsymbol{u}$.

The true Q value function is defined as the expectation of accumulative rewards, i.e., $\mathcal{Q}(s_t, \boldsymbol{u}_t) := \mathbb{E}_{s_{t+1:\infty}, \boldsymbol{u}_{t+1:\infty}}[R_t|s_t, \boldsymbol{u}_t]$, where $R_t = \sum_{i=0}^{\infty} \gamma^i r_{t+1}$. $\mathcal{Q}(s_t, \boldsymbol{u}_t)$ is approximated by the joint Q value function $Q(s, \boldsymbol{u})$. We denote the optimal action and greedy action by $\boldsymbol{u}^* := argmax_{\boldsymbol{u}} \mathcal{Q}(s, \boldsymbol{u})$ and $\boldsymbol{u}_{gre} := argmax_{\boldsymbol{u}} Q(s, \boldsymbol{u})$, respectively.

### 2.2 VALUE FACTORIZATION

In value factorization, the joint Q value function is factorized through a value factorization operator $\mathcal{F}(\cdot)$ as

$$Q(s, \boldsymbol{u}) = \mathcal{F}(Q_1(\tau_1, u_1), \cdots, Q_n(\tau_n, u_n)) \tag{1}$$

$Q_i(u_i, \tau_i) : U_i \to \mathcal{R}$ $(i \in [1, n])$ is defined as the local Q value function of agent $i$. A critical rule of value factorization is the Independent Global Max principle. The IGM principle is defined as the identity of the joint greedy action and the set of local greedy actions. Formally, given the joint Q

value function $Q(s, \boldsymbol{u})$ and the factorized local Q functions $\{Q_1(\tau_1, u_1), \cdots, Q_n(\tau_n, u_n)\}$ by $\mathcal{F}(\cdot)$, if the following equality holds

$$argmax_{\boldsymbol{u}} \, Q(s, \boldsymbol{u}) = \{argmax_{u^1} \, Q_1(\tau_1, u_1), \cdots, argmax_{u_n} \, Q_n(\tau_n, u_n)\} \qquad (2)$$

we say the factorization operator satisfies the IGM principle. The IGM principle enables the coordination of local policies under the centralized trained joint Q value function.

Linear Value Factorization (LVF) naturally meets the IGM principle and becomes the most popular value factorization method in recent years. In LVF, the joint Q value function is linearly factorized as

$$Q(s, \boldsymbol{u}) = \mathcal{F}(Q_1(\tau_1, u_1), \cdots, Q_n(\tau_n, u_n)) = \sum_{i=1}^{n} w_i Q_i(\tau_i, u_i) + V(s) \qquad (3)$$

The joint Q value function of LVF can only represent linearly factorizable true Q value functions, known as the problem of representation limitation. As a result, the optimal Bellman operator could be not a $\gamma-$constraint (Wang et al., 2020a) when faced with non-linear true Q value functions. In words, there could be multiple convergences for the joint Q value function (Wan et al., 2021) and the policy would get trapped in sub-optimums.

## 3 INVESTIGATION OF LINEAR VALUE FACTORIZATION IN MARKOV GAMES

### 3.1 DECOMPOSABILITY OF MARKOV GAMES

The linearly factorizable joint Q value function in LVF is incapable to represent non-linear true Q value functions. In this section, we investigate the conditions of the linearity of the true Q value function in the context of the Markov game. A Markov game (Littman, 1994) is equivalent to a decentralized fully observable Markov decision process, which can described by a tuple $\mathcal{MG} = <\boldsymbol{\mathcal{S}}, \boldsymbol{\mathcal{U}}, \mathbb{P}, \boldsymbol{r}, n, \gamma>$. The explanation of the symbols can be found in the preliminary. Firstly, we introduce the concept of decomposability of Markov games.

**Definition 1 (Decomposable Markov Game).** Given an Markov game (Dou et al., 2022) $\mathcal{MG} = <\boldsymbol{\mathcal{S}}, \boldsymbol{\mathcal{U}}, \mathcal{P}, r, n, \gamma>$, if there exists a collection of subspaces of the joint state-action space $\{\mathcal{S}_1 \times \hat{\mathcal{U}}_1, \mathcal{S}_2 \times \hat{\mathcal{U}}_2, \cdots, \mathcal{S}_k \times \hat{\mathcal{U}}_k\}$ $(k \geq 2)$, i.e., $\mathcal{S}_i \times \hat{\mathcal{U}}_i \subset \boldsymbol{\mathcal{S}} \times \boldsymbol{\mathcal{U}}$ for $\forall i \in [1, k]$, such that for $\forall (\boldsymbol{s}_t, \boldsymbol{u}_t) \in \boldsymbol{\mathcal{S}} \times \boldsymbol{\mathcal{U}}$, the following holds

- $\forall i \in [1, k], \mathcal{P}(s_{i,t+1}|s_{i,t}, \hat{u}_{i,t}) = \mathcal{P}(s_{i,t+1}|\boldsymbol{s}_t, \boldsymbol{u}_t)$, where $(s_{i,t}, \hat{u}_{i,t}) \in \mathcal{S}_i \times \hat{\mathcal{U}}_i$;

- the reward function $\boldsymbol{r}(\boldsymbol{s}_t, \boldsymbol{u}_t)$ can be linearly factorized as $\boldsymbol{r}(\boldsymbol{s}_t, \boldsymbol{u}_t) = \sum_{i=1}^{k} r_i(s_{i,t}, \hat{u}_{i,t})$.

then we say that $\mathcal{MG}$ is decomposable by $\{\mathcal{MG}_1, \mathcal{MG}_2, \cdots, \mathcal{MG}_k\}$, where $\mathcal{MG}_i := <\mathcal{S}_i, \hat{\mathcal{U}}_i, \mathcal{P}, r_i, n_i, \gamma>$ $(i \in [1, k])$. Otherwise we say $\mathcal{MG}$ is indecomposable. $n_i$ is the number of agents involved in $\mathcal{MG}_i$. Specially, if $\mathcal{MG}_i$ is no longer decomposable for $\forall i \in [1, k]$, we say that $\{\mathcal{MG}_1, \mathcal{MG}_2, \cdots, \mathcal{MG}_k\}$ is the Minimum Granularity Decomposition (MGD).

$\mathcal{MG}_i, \mathcal{MG}_j$ $(\forall i, j \in [1, k]$ and $i \neq j)$ should not be considered as elements of the decomposition in the following situations. (1) Void decomposition: $\hat{\mathcal{U}}_i = \emptyset$ and $r_i(s_i, \hat{u}_i) = 0$; (2) Self-decomposition: $\hat{\mathcal{U}}_i = \boldsymbol{\mathcal{U}}$ and $r_i(s_i, \hat{u}_i) = C \cdot \boldsymbol{r}(\boldsymbol{s}, \boldsymbol{u})$, where $C$ is a constant; (3) Overlapping decomposition: $\hat{\mathcal{U}}_i = \hat{\mathcal{U}}_j$ and $r_i(s_i, \hat{u}_i) = C \cdot r_j(s_j, \hat{u}_j)$. Therefore, we also require $\forall i, j \in [1, k]$ $(i \neq j), \hat{\mathcal{U}}_i \neq \emptyset, \hat{\mathcal{U}}_i \neq \boldsymbol{\mathcal{U}}$ and $\hat{\mathcal{U}}_i \neq \hat{\mathcal{U}}_j$ for a decomposable Markov game.

Examples of both decomposable and indecomposable Markov games are provided in Fig.3.1, where 4 agents (denoted by dots) need to cover 2 landmarks (denoted by squares) in pairs. Agents are assigned target landmarks in colors. The team receives an instant reward when any agent covers the target landmark. In the indecomposable case, the team only receives a reward when a landmark is covered by the first 2 agents, where the reward function is not linearly factorizable since it is determined by the policy of all agents. Fig.3.1 (c) and (e) present two decompositions of the decomposable Markov game. Especially, the decomposition in Fig.3.1(e) is the MGD since none of the decomposed Markov games are further decomposable.

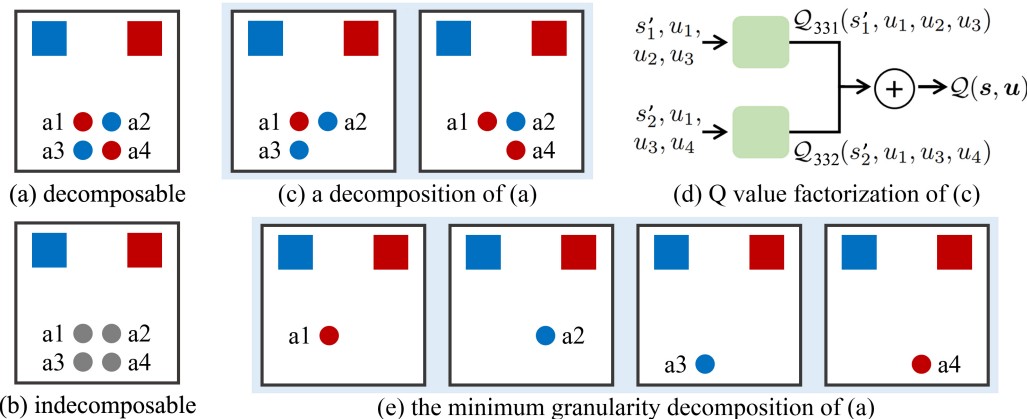

Figure 1: Examples of decomposable and indecomposable Markov games.

**Proposition 1 (Linear factorizability of the true Q value function in decomposable Markov games).** The true Q value function can be linearly factorized as $\mathcal{Q}(\boldsymbol{s}, \boldsymbol{u}) = \sum_{i=1}^{k} \mathcal{Q}_i(s_i, \hat{u}_i)$ for $\forall (\boldsymbol{s}, \boldsymbol{u}) \in \boldsymbol{\mathcal{S}} \times \boldsymbol{\mathcal{U}}$ *if and only if* $\mathcal{MG}$ *is decomposable by* $\{\mathcal{MG}_1, \mathcal{MG}_2, \cdots, \mathcal{MG}_k\}$.

The proof of Proposition 1 can be found in Appendix A. Fig.3.1(d) presents the factorization of the true Q value function under the decomposition in Fig.3.1(c). The joint Q value function of LVF is capable to represent the true Q value function only if each decomposed Markov game of the MGD involves only a single agent. Note that the decomposition of a Markov game is non-unique. We can obtain new decompositions from an existing decomposition, for which we introduce the following lemma:

**Lemma 1.** Suppose $\{\mathcal{MG}_1, \mathcal{MG}_2, \cdots, \mathcal{MG}_k\}$ $(k \geq 2)$ is a decomposition of Markov game $\mathcal{MG}$. $\{\mathcal{MG}'_1, \mathcal{MG}'_2, \cdots, \mathcal{MG}'_{k_s}\}$ is also a decomposition of $\mathcal{MG}$ if the following conditions holds: (1) $\mathcal{MG}'_j$ is decomposable by a non-empty subset of $\{\mathcal{MG}_1, \mathcal{MG}_2, \cdots, \mathcal{MG}_k\}$ for $\forall j \in [1, k_s]$; (2) $\cup_{j=1}^{k_s} \{\mathcal{S}'_j \times \hat{\mathcal{U}}'_j\} = \boldsymbol{\mathcal{S}} \times \boldsymbol{\mathcal{U}}$, where $\mathcal{MG}'_j = <\mathcal{S}'_j, \mathcal{U}'_j, \mathcal{P}, r'_j, n'_j, \gamma>$.

The proof of Lemma 1 can be found in Appendix B. Obviously, if $\{\mathcal{MG}_1, \mathcal{MG}_2, \cdots, \mathcal{MG}_k\}$ $(k \geq 2)$ is a decomposition of Markov game $\mathcal{MG}$, we can also obtain new decompositions by further decomposing the elements of $\{\mathcal{MG}_1, \mathcal{MG}_2, \cdots, \mathcal{MG}_k\}$.

## 3.2 LVF in Indecomposable Markov Games

Decomposability is unusual for Markov games. Multi-agent tasks involving cooperative rewards or interactive transitions of all agents are usually indecomposable Markov games, where the true Q value functions are not linearly factorizable. In this subsection, we investigate the performance of LVF in the most frequent cases, i.e., indecomposable Markov games. Our investigation is carried out from the perspective of indecomposable Markov games with discrete action spaces, where the representation of the true Q value function is equivalent to solving the linear equation system

$$\left\{ \mathcal{Q}(\boldsymbol{s}, \boldsymbol{u}) = \sum_{i=1}^{n} w_i Q_i(\boldsymbol{s}, u_i) + V(\boldsymbol{s}) \right\}_{\forall \boldsymbol{u} \in \boldsymbol{\mathcal{U}}} \tag{4}$$

The maximum number of independent equations is $m^n$, where $m$ is the size of the discrete action space. It can be proved that the rank of the coefficient matrix of the equation system equals $n(m - 1) + 1$ (the proof is available in Appendix C). The equation system is overdetermined since $m^n > n(m-1)+1$ for $\forall m, n \in [2, \infty)$. Despite the representation error of the joint Q value function, the target of the joint Q value function is always unbiased for LVF under value iteration in sarsa manner. To explain this, we introduce the following proposition:

**Proposition 2.** In indecomposable Markov game, the estimate of state value function is unbiased for LVF under the value iteration in sarsa manner, i.e., $\sum_{\boldsymbol{u}}^{m^n} \pi(\boldsymbol{u}|\boldsymbol{s}) Q(\boldsymbol{s}, \boldsymbol{u}) = \sum_{\boldsymbol{u}}^{m^n} \pi(\boldsymbol{u}|\boldsymbol{s}) \mathcal{Q}(\boldsymbol{s}, \boldsymbol{u})$.

The proof of Proposition 2 can be found in Appendix D. Furthermore, we have

$$Q_{target,t} = r(\boldsymbol{s}_t, \boldsymbol{u}_t) + \gamma \frac{1}{m^n} \sum_{\boldsymbol{u}_{t+1}}^{m^n} \int_{\boldsymbol{s}_{t+1}} \mathbb{P}(\boldsymbol{s}_{t+1}|\boldsymbol{s}_t, \boldsymbol{u}_t) \pi(\boldsymbol{u}_{t+1}|\boldsymbol{s}_{t+1}) Q(\boldsymbol{s}_{t+1}, \boldsymbol{u}_{t+1}) d\boldsymbol{s}_{t+1}$$

$$= r(\boldsymbol{s}_t, \boldsymbol{u}_t) + \gamma \mathbb{E}_{\boldsymbol{u}_{t+1}\sim\pi(\boldsymbol{u}_{t+1}|\boldsymbol{s}_{t+1}), \boldsymbol{s}_{t+1}\sim\mathbb{P}(\boldsymbol{s}_{t+1}|\boldsymbol{s}_t, \boldsymbol{u}_t)} \left[ Q(\boldsymbol{s}_{t+1}, \boldsymbol{u}_{t+1}) \right] = \mathcal{Q}(\boldsymbol{s}_t, \boldsymbol{u}_t) \tag{5}$$

Eq.5 indicates the target of the joint Q value function still equals the true Q value function in inde-composable Markov games for LVF under the value iteration in sarsa manner. In this case, LVF is capable to find the optimal policy if for $\forall t \in [0, \infty)$, the following holds

$$\boldsymbol{u}_t^* = \underset{\boldsymbol{u}}{argmax} \, \mathcal{Q}_{LVF}(\boldsymbol{s}_t^*, \boldsymbol{u}_t) \tag{6}$$

where $\boldsymbol{\tau}^* := (\boldsymbol{s}_0^*, \boldsymbol{u}_0^*, \boldsymbol{s}_1^*, \boldsymbol{u}_1^*, \cdots)$ is the optimal trajectory, i.e., $\forall t \in [0, \infty)$ $\boldsymbol{u}_t^* = argmax_{\boldsymbol{u}} \, \mathcal{Q}(\boldsymbol{s}_t^*, \boldsymbol{u}_t)$. Eq.6 is equivalent to solving a single-step matrix game. But note that the joint Q value function is a biased estimate of the true Q value function. Therefore, we have $max_{\boldsymbol{u}_t} Q(\boldsymbol{s}_t, \boldsymbol{u}_t) \neq max_{\boldsymbol{u}_t} \mathcal{Q}(\boldsymbol{s}_t, \boldsymbol{u}_t)$, which suggests there are errors in the Q-learning target. Such errors could accumulate alone the trajectories by the bootstrap of the joint Q value function.

## 4 VALUE FACTORIZATION FUNCTIONS FOR INDECOMPOSABLE MARKOV

Although the target of the joint Q value function exactly equals the true Q value function in inde-composable Markov games for LVF under the value iteration in sarsa manner, it is still impractical for LVF to solve every single step matrix game in the optimal trajectory. To deal with the indecom-posable Markov game, in this section, we turn to value factorization functions that satisfy both IGM and CRC conditions. According to the partial derivative on local Q value functions, we divide the value factorization functions into linear and non-linear.

### 4.1 EXTEND LINEAR VALUE FACTORIZATION FUNCTION

Firstly, consider a linear value factorization function $\mathcal{F}(Q_1(\tau_1, u_1), \cdots, Q_n(\tau_n, u_n))$. Let $Q_{set}(\boldsymbol{\tau}, \boldsymbol{u}) := \{Q_1(\tau_1, u_1), \cdots, Q_n(\tau_n, u_n)\}$ denote the collection of local Q value functions. We have $\partial \mathcal{F}(Q_{set}(\boldsymbol{\tau}, \boldsymbol{u}))/\partial Q_i(\tau_i, u_i) = w_i$ for $\forall i \in [1, n]$. To improve the representation capability of $\mathcal{F}(\cdot)$, we introduce a set of parameterized modules denoted by $\mathcal{M}_{set}(\boldsymbol{s}, \boldsymbol{u}) := \{\mathcal{M}_1(s_1, \hat{u}_1), \cdots, \mathcal{M}_k(s_k, \hat{u}_k)\}$. $(s_i, \hat{u}_i) \in \mathcal{S}_i \times \hat{\mathcal{U}}_i \, (i \in [1, k])$, where $\mathcal{S}_i \times \hat{\mathcal{U}}_i \subset \boldsymbol{\mathcal{S}} \times \boldsymbol{\mathcal{U}}$. The joint Q value function equals

$$Q(\boldsymbol{\tau}, \boldsymbol{u}) = \mathcal{F}(Q_{set}(\boldsymbol{\tau}, \boldsymbol{u}), \mathcal{M}_{set}(\boldsymbol{s}, \boldsymbol{u})) = \sum_{i=1}^{n} w_i Q_i(\tau_i, u_i) + \sum_{j=1}^{k} \mathcal{M}_j(s_j, \hat{u}_j) + V(\boldsymbol{s}) \tag{7}$$

To distinguish from LVF in Eq.1, we refer to the function in Eq.7 as extended LVF. Note that inde-composable Markov games are not decomposable on any collection of subspaces of the joint state-action space. According to Proposition 1, the true Q value function is also not linearly factorizable by any functions based on the proper subspaces of the joint state-action space. In words, a neces-sary condition for Eq.7 to represent any true Q value functions is $\exists \mathcal{M}(s_j, \hat{u}_j), (s_j, \hat{u}_j) = (\boldsymbol{s}, \boldsymbol{u})$ $(j \in [1, k])$. Notice $\mathcal{F} : \boldsymbol{\mathcal{S}} \times \boldsymbol{\mathcal{U}} \to (-\infty, \mathcal{Q}^*]$. Therefore, we also require $\mathcal{M}_j : \boldsymbol{\mathcal{S}} \times \boldsymbol{\mathcal{U}} \to (-\infty, C]$, where $C$ is an arbitrary constant. Now we consider the IGM principle. The IGM principle requires $\partial \mathcal{F}(Q_{set}(\boldsymbol{\tau}, \boldsymbol{u}), \mathcal{M}_{set}(\boldsymbol{s}, \boldsymbol{u}))/\partial Q_i(\tau_i, u_i) = w_i > 0$ and $\mathcal{M}_j(s_j, \hat{u}_j) \leq \mathcal{M}_j(s_j, \hat{u}_{j,gre})$. Based on the constraints above, we list some examples of extended LVF functions, which is available in Appendix.E.

### 4.2 NON-LINEAR VALUE FACTORIZATION FUNCTION

Linear value factorization functions constitute a small part of the whole value factorization function family. In this subsection, we discuss the functions of Non-linear Value Factorization (NVF). We have $\partial \mathcal{F}(Q_{set}(\boldsymbol{\tau}, \boldsymbol{u}))/\partial Q_i(\tau_i, u_i) = f_i(Q_{set}(\boldsymbol{\tau}, \boldsymbol{u}), s_i, \hat{u}_i)$ for NVF functions, where $\mathcal{S}_i \times \hat{\mathcal{U}}_i \subset \boldsymbol{\mathcal{S}} \times \boldsymbol{\mathcal{U}}$. There are two different approaches to improve the representation capability of the function: (1) Introducing parameterized functions directly; (2) introducing parameterized modules.

Let $\mathcal{F}_\theta(Q_{set}(\boldsymbol{\tau}, \boldsymbol{u}))$ denote the parameterized functions, where $\theta$ is the collection of introduced parameters. A defect of $\mathcal{F}_\theta(Q_{set}(\boldsymbol{\tau}, \boldsymbol{u}))$ is the uncontrollable sign of the derivative of local Q value functions. As a result, the function suffers from poor convergence. More details are provided in Appendix E.

Consider the second approach, i.e., introducing parameterized modules in a predefined NVF. Let $\mathcal{M}_{set}(\boldsymbol{s}, \boldsymbol{u}) := \{\mathcal{M}_1(s_1, \hat{u}_1), \cdots, \mathcal{M}_k(s_k, \hat{u}_k)\}$ denote the introduced modules, we have $Q(\boldsymbol{\tau}, \boldsymbol{u}) = \mathcal{F}(Q_{set}(\boldsymbol{\tau}, \boldsymbol{u}), \mathcal{M}_{set}(\boldsymbol{s}, \boldsymbol{u}))$. We denote the partial derivatives of $Q_i(\tau_i, u_i)$ by $\mathcal{F}_i' := \partial \mathcal{F}(Q_{set}(\boldsymbol{\tau}, \boldsymbol{u}), \mathcal{M}_{set}(\boldsymbol{s}, \boldsymbol{u})) / \partial Q_i(\tau_i, u_i)$ $(i \in [1, n])$. For good convergence, we expect that $\mathcal{F}_i' > 0$ for $Q_i(\tau_i, u_i) \in (-\infty, Q_i(\tau_i, u_{i,gre})]$, which is a more strict constraint than the IGM principle. Since the function $\mathcal{F}(\cdot)$ is predefined with a fixed form, a necessary condition of CRC is $\exists \mathcal{M}(s_j, \hat{u}_j), (s_j, \hat{u}_j) = (\boldsymbol{s}, \boldsymbol{u})$ $(j \in [1, k])$. Based on the constraints above, we list some examples of NVF function, which is available in Appendix.E.

# 5 METHODOLOGY

## 5.1 REPRESENTATION INTERFERENCE

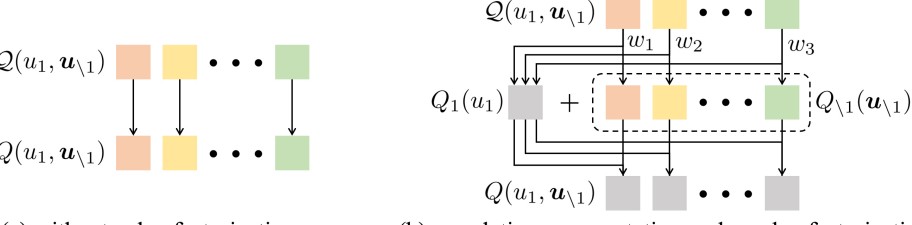

(a) without value factorization          (b) correlation representation under value factorization

Figure 2: In value factorization, the representation of all true Q values involving $u_1$ is interfered by each other through the training of the shared local Q value function $Q_1(u_1)$.

A common problem of value decomposition is representation interference among true Q values. As shown in Fig.5.1, the local Q value function $Q_i(\tau_i, u_i)$ is an input shared by all joint Q values involving $u_i$ $(i \in [1, n])$. Since the joint Q values are representations of corresponding true Q values, the representations of all true Q values involving $u_i$ are correlated via the training of $Q_i(\tau_i, u_i)$. Methods that ignore the correlation of representation would suffer from poor convergence, where the gradient on $Q_i(\tau_i, u_i^*)$ contributed by the optimal true Q value is interfered or even submerged in the gradients contributed by the correlated non-optimal true Q values. In words, the representation of $\mathcal{Q}(\boldsymbol{s}, \boldsymbol{u}^*)$ is interfered by its representation correlation with non-optimal true Q values. Let $w_i^*$ denote the relative weight of gradient contributed by $\mathcal{Q}(\boldsymbol{\tau}, \boldsymbol{u}^*)$ in all true Q values involving $u_i$, i.e.,

$$w_i^* = \frac{\pi(\boldsymbol{u}^*|\boldsymbol{s}) \cdot \frac{\partial \mathcal{F}}{\partial Q_i}|_{\boldsymbol{u}=\boldsymbol{u}^*}}{\sum_{u_{\backslash i}}^{\mathcal{U}^{n-1}} \pi(u_i^*, u_{\backslash i}|\boldsymbol{s}) \cdot \frac{\partial \mathcal{F}}{\partial Q_i}|_{\boldsymbol{u}=\{u_i^*, u_{\backslash i}\}}} \tag{8}$$

where $u_{\backslash i}$ denote the joint action of all agents except $i$. The representation interference on $\mathcal{Q}(\boldsymbol{\tau}, \boldsymbol{u}^*)$ is negatively correlated to $w_i^*$. For linear value factorization function, according to Eq.7, we have $\forall i \in [1, n]$, $\frac{\partial \mathcal{F}}{\partial Q_i}|_{\forall \boldsymbol{u} \in \mathcal{U}^n} = w_i$, where $w_i^* = \frac{\pi(\boldsymbol{u}^*|\boldsymbol{s})}{\sum_{u_{\backslash i}}^{\mathcal{U}^{n-1}} \pi(u_i^*, u_{\backslash i}|\boldsymbol{s})}$ is mainly determined by the sample distribution. By contrast, for non-linear value factorization function, $\frac{\partial \mathcal{F}}{\partial Q_i}$ is a function of $\boldsymbol{u}$. An example is QPLEX, where the representation interference is serious due to the sharply decreasing $w_i^*$ during training. More details and analysis of QPLEX can be found in Appendix F. Since the representation interference on $\mathcal{Q}(\boldsymbol{\tau}, \boldsymbol{u}^*)$ is related to the form of the value factorization function, we consider to design a non-linear value factorization function to address the problem.

## 5.2 REPRESENTATION INTERFERENCE SUPPRESSION VIA NON-LINEAR VALUE FACTORIZATION

To alleviate the representation interference on $\mathcal{Q}(\boldsymbol{u}^*)$, we consider raising the relative weight of gradient contributed by $\mathcal{Q}(\boldsymbol{u}^*)$, i.e., $w_i^*$. Referring to Eq.8, $w_i^*$ is determined by the sample distribution and the partial derivatives of $\mathcal{F}(Q_i(u_i^*), Q_{\backslash i}(u_{\backslash i}))$ with respect to $Q_i(u_i^*)$, where $Q_{\backslash i}(u_{\backslash i})$ denotes the set of all agents' local Q value functions except $u_i$. Note that $w_i^*$ continuously reduces during the training in QPLEX, where the value factorization function is

$$\mathcal{F}(Q_{set}(\boldsymbol{\tau}, \boldsymbol{u}), \mathcal{M}_{set}(\boldsymbol{s}, \boldsymbol{u})) = -\sum_{i=1}^{n} |\mathcal{M}_i(\boldsymbol{s}, \boldsymbol{u})| \cdot [Q_i(\tau_i, u_{i,gre}) - Q_i(\tau_i, u_i)] + \sum_{i=1}^{n} Q_i(\tau_i, u_{i,gre})$$
$$(9)$$

We make a slight change on the function to reverse the trend.

$$\mathcal{F}(Q_{set}(\boldsymbol{\tau}, \boldsymbol{u}), \mathcal{M}_{set}(\boldsymbol{s}, \boldsymbol{u}))$$
$$= -\sum_{i=1}^{n} \left[ Q_i(\tau_i, u_{i,gre}) - e^{-I(\boldsymbol{u}=\boldsymbol{u}_{gre}) \cdot |\mathcal{M}_i(\boldsymbol{s}, \boldsymbol{u})|} \cdot Q_i(\tau_i, u_i) \right] + \sum_{i=1}^{n} Q_i(\tau_i, u_{i,gre})$$
$$= \sum_{i=1}^{n} e^{-I(\boldsymbol{u}=\boldsymbol{u}_{gre}) \cdot |\mathcal{M}_i(\boldsymbol{s}, \boldsymbol{u})|} \cdot Q_i(\tau_i, u_i)$$
$$(10)$$

where $I(\boldsymbol{u} = \boldsymbol{u}_{gre})$ equals 1 if $\boldsymbol{u} = \boldsymbol{u}_{gre}$ otherwise 0. When $Q_i(\tau_i, u_i) > 0$, we have $\partial \mathcal{F}/\partial |\mathcal{M}_i(\boldsymbol{s}, \boldsymbol{u})| = -e^{-I(\boldsymbol{u}=\boldsymbol{u}_{gre}) \cdot |\mathcal{M}_i(\boldsymbol{s}, \boldsymbol{u})|} Q_i(\tau_i, u_i) < 0$, i.e., $|\mathcal{M}_i(\boldsymbol{s}, \boldsymbol{u})|$ decreases as $Q(\boldsymbol{\tau}, \boldsymbol{u}) = \mathcal{F}(Q_{set}(\boldsymbol{\tau}, \boldsymbol{u}), \mathcal{M}_{set}(\boldsymbol{s}, \boldsymbol{u}))$ grows. Let $\mathcal{F}_i' := \partial \mathcal{F}(Q_{set}(\boldsymbol{u}), \mathcal{M}_{set}(\boldsymbol{u}))/\partial Q_i(\tau_i, u_i)$ denote the partial derivative of $Q_i(\tau_i, u_i)$. We have $\mathcal{F}_i' = e^{-I(\boldsymbol{u}=\boldsymbol{u}_{gre}) \cdot |\mathcal{M}_i(\boldsymbol{s}, \boldsymbol{u})|}$, which is negatively related to $|\mathcal{M}_i(\boldsymbol{s}, \boldsymbol{u})|$. Therefore, $\mathcal{F}_i'$ is positively related to $Q(\boldsymbol{\tau}, \boldsymbol{u})$ when $Q_i(\tau_i, u_i) > 0$. To ensure $Q_i(\tau_i, u_i) > 0$, we replace $Q_i(\tau_i, u_i)$ with $|Q_i(\tau_i, u_i)|$.

$$\mathcal{F}(Q_{set}(\boldsymbol{\tau}, \boldsymbol{u}), \mathcal{M}_{set}(\boldsymbol{s}, \boldsymbol{u})) = \sum_{i=1}^{n} e^{-I(\boldsymbol{u}=\boldsymbol{u}_{gre}) \cdot |\mathcal{M}_i(\boldsymbol{s}, \boldsymbol{u})|} \cdot |Q_i(\tau_i, u_i)| + V(\boldsymbol{s})$$
$$(11)$$

where $V(\boldsymbol{s})$ enables $\mathcal{F}(Q_{set}(\boldsymbol{\tau}, \boldsymbol{u}), \mathcal{M}_{set}(\boldsymbol{s}, \boldsymbol{u}))$ to represent negative true Q values. Based on the value factorization function above, we introduce our method, namely, Q Factorization with Representation Interference Suppression (QFRIS). The value factorization function of QFRIS equals

$$\mathcal{F}_{QFRIS}(Q_{set}(\boldsymbol{\tau}, \boldsymbol{u}), \mathcal{M}_{set}(\boldsymbol{s}, \boldsymbol{u}))$$
$$= e^{-|\mathcal{M}(\boldsymbol{s}, \boldsymbol{u})|^2 \cdot I(\boldsymbol{u}=\boldsymbol{u}_{gre})} \sum_{i=1}^{n} Q_i(\tau_i, u_i) - |\mathcal{M}(\boldsymbol{s}, \boldsymbol{u})| \cdot I(\boldsymbol{u} = \boldsymbol{u}_{gre}) + V(\boldsymbol{s})$$
$$(12)$$

Obviously, our QFRIS satisfies both IGM and CRC conditions. The network structure of QFRIS is provided in Appendix G.

## 6 EXPERIMENTS

Our experiments consist of 4 parts. Firstly, we verify our propositions on a finite Markov Game; Secondly, we compare the performance of QFRIS value factorization with the value factorization functions in other methods. Finally, we evaluate the performance of QFRIS on predator-pery and StarCraft Multi-Agent Challenge (SMAC). The latter three parts are available in Appendix H.

We design toy games for both decomposable and indecomposable cases of Fig.3.1 and carry out experiments to verify our propositions about the decomposability of Markov games. The tasks are shown in Fig.6(a), where 4 agents (denoted by dots) need to cover 2 landmarks (denoted by squares) in pairs. The map is gridded in a $4 \times 4$ checkerboard. All agents are initialized with the position $(3, 0)$ and required to select actions from $\{up, right\}$ at each time step. Each agent is assigned with target landmark in color. The team receives an instant reward of 1.0 when any agent covers the target landmark. For the indecomposable case, the team only receives reward when a landmark is covered by the first 2 agents. The invalid actions, e.g., $up$ at position $(0, 0)$ are masked.

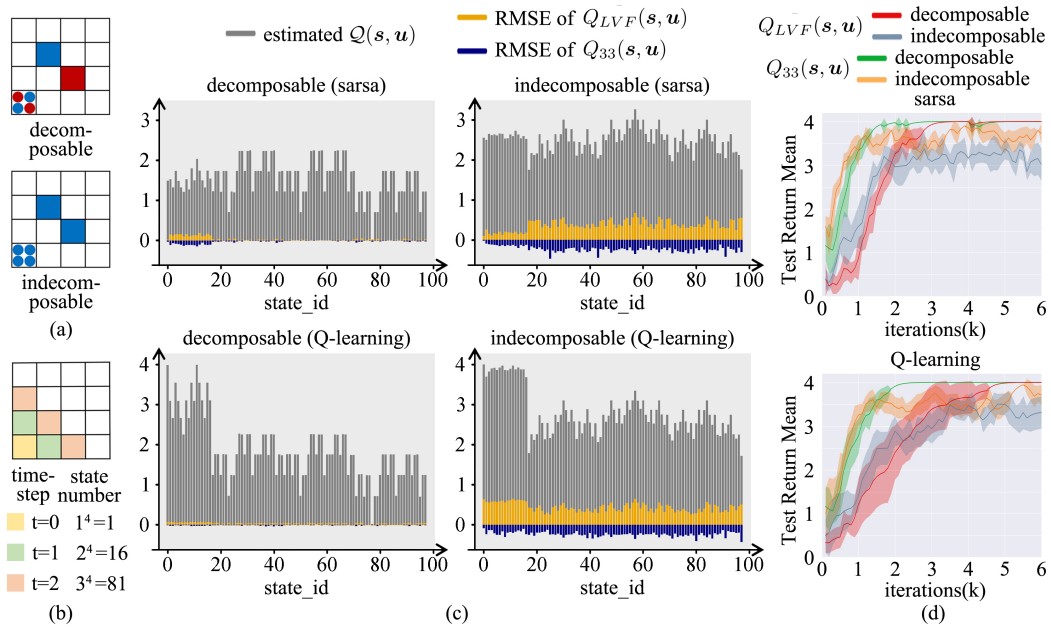

Figure 3: Verification of the factorizability of the true Q value functions in decomposable & indecomposable Markov games. (a) Tasks for decomposable & indecomposable cases; (b) state number in first 3 time steps; (c) RMSE of linearly factorized joint Q value functions; (d) test mean return of linearly factorized joint Q value functions.

**Verification of proposition 1**. For the decomposable case, two decompositions of $\mathcal{MG}$ are shown in Fig.3.1(c) and Fig.3.1(e). According to proposition 1, the true Q value function can be linearly factorized as $\mathcal{Q}(\boldsymbol{s}, \boldsymbol{u}) = \mathcal{Q}_{331}(s'_1, u_1, u_2, u_3) + \mathcal{Q}_{332}(s'_2, u_1, u_2, u_4)$ or $\mathcal{Q}(\boldsymbol{s}, \boldsymbol{u}) = \mathcal{Q}_1(s_1, u_1) + \mathcal{Q}_2(s_2, u_2) + \mathcal{Q}_2(s_3, u_3) + \mathcal{Q}_2(s_4, u_4)$. We apply two neural networks denoted by $Q_{33}(\boldsymbol{s}, \boldsymbol{u})$ and $Q_{MGD}(\boldsymbol{s}, \boldsymbol{u})$ to model $\mathcal{Q}(\boldsymbol{s}, \boldsymbol{u})$, respectively. Each network sums the approximated true Q value function of decomposed Markov games, e.g., 3.1.(d) for $Q_{33}(\boldsymbol{s}, \boldsymbol{u})$. To verify the factorizability of $\mathcal{Q}(\boldsymbol{s}, \boldsymbol{u})$, we evaluate the estimated error of $Q_{33}(\boldsymbol{s}, \boldsymbol{u})$ and $Q_{MGD}(\boldsymbol{s}, \boldsymbol{u})$. To be specific, we approximate $\mathcal{Q}(\boldsymbol{s}, \boldsymbol{u})$ by a non-factorized neural network denoted by $Q(\boldsymbol{s}, \boldsymbol{u})$. As shown in Fig.6(b), based on the positions of all agents, there are totally 98 states in the first 3 time steps. At each state, we calculate the Root Mean Square Error (RMSE) of $Q_{33}(\boldsymbol{s}, \boldsymbol{u})$ and $Q_{MGD}(\boldsymbol{s}, \boldsymbol{u})$, respectively, e.g.,

$$RMSE_{33}(\boldsymbol{s}) = \left( \frac{1}{m^n} \sum_{\boldsymbol{u}}^{\mathcal{U}^n} [Q(\boldsymbol{s}, \boldsymbol{u}) - Q_{33}(\boldsymbol{s}, \boldsymbol{u})]^2 \right)^{\frac{1}{2}} \tag{13}$$

All agents follow random policies. The experimental results after 6k steps of training are shown in Fig.6(c), where each bar denotes the result of a single state. The estimation errors of $Q_{33}(\boldsymbol{s}, \boldsymbol{u})$ and $Q_{MGD}(\boldsymbol{s}, \boldsymbol{u})$ are negligible for the decomposable case but sizable for the indecomposable case, which suggests $\mathcal{Q}(\boldsymbol{s}, \boldsymbol{u})$ is linearly factorizable only if the Markov game is decomposable. We also test the return under $Q_{33}(\boldsymbol{s}, \boldsymbol{u})$ and $Q_{MGD}(\boldsymbol{s}, \boldsymbol{u})$ in both decomposable and indecomposable cases. The results are shown in Fig.6(d). The task is solved when all agents cover the target landmarks, i.e., the return equals 4.0. Both $Q_{33}(\boldsymbol{s}, \boldsymbol{u})$ and $Q_{MGD}(\boldsymbol{s}, \boldsymbol{u})$ are able to handle the decomposable case. But for the indecomposable case, both joint Q value functions fail to solve the task since the true Q value function is not linearly factorizable.

**Verification of Proposition 2**. According to proposition 2, the estimate of state value function is unbiased for LVF under the value iteration in sarsa manner. We model the linearly factorized $\mathcal{Q}(\boldsymbol{s}, \boldsymbol{u})$ under MGD and the non-factorized $\mathcal{Q}(\boldsymbol{s}, \boldsymbol{u})$ by $Q_{MGD}(\boldsymbol{s}, \boldsymbol{u})$ and $Q(\boldsymbol{s}, \boldsymbol{u})$, respectively. The state value function can be approximated by $V(\boldsymbol{s}) = \sum_{\boldsymbol{u}}^{\mathcal{U}^n} \pi(\boldsymbol{u}|\boldsymbol{s})Q(\boldsymbol{s}, \boldsymbol{u})$. The estimated state value function of LVF equals $V_{MGD}(\boldsymbol{s}) = \sum_{\boldsymbol{u}}^{\mathcal{U}^n} \pi(\boldsymbol{u}|\boldsymbol{s})Q_{MGD}(\boldsymbol{s}, \boldsymbol{u})$. Note that the target of $Q(\boldsymbol{s}_t, \boldsymbol{u}_t)$ equals $r(\boldsymbol{s}_t, \boldsymbol{u}_t) + \gamma P(\boldsymbol{s}_{t+1}|\boldsymbol{s}_t, \boldsymbol{u}_t)V(\boldsymbol{s}_{t+1})$ for sarsa. To evaluate the

error of the representation target for LVF in indecomposable Markov games, we calculate the difference between $V(\boldsymbol{s})$ and $V_{MGD}(\boldsymbol{s})$. Besides, the target of $Q(\boldsymbol{s}_t, \boldsymbol{u}_t)$ equals $r(\boldsymbol{s}_t, \boldsymbol{u}_t) + \gamma P(\boldsymbol{s}_{t+1}|\boldsymbol{s}_t, \boldsymbol{u}_t)max_{\boldsymbol{u}_{t+1}}Q(\boldsymbol{s}_{t+1}, \boldsymbol{u}_{t+1})$ for Q-learning. We also calculate the difference between $max_{\boldsymbol{u}_t}Q(\boldsymbol{s}_t, \boldsymbol{u}_t)$ and $max_{\boldsymbol{u}_t}Q_{MGD}(\boldsymbol{s}_t, \boldsymbol{u}_t)$. The experimental results are shown in Fig.6, where each bar denotes the result of a single state.

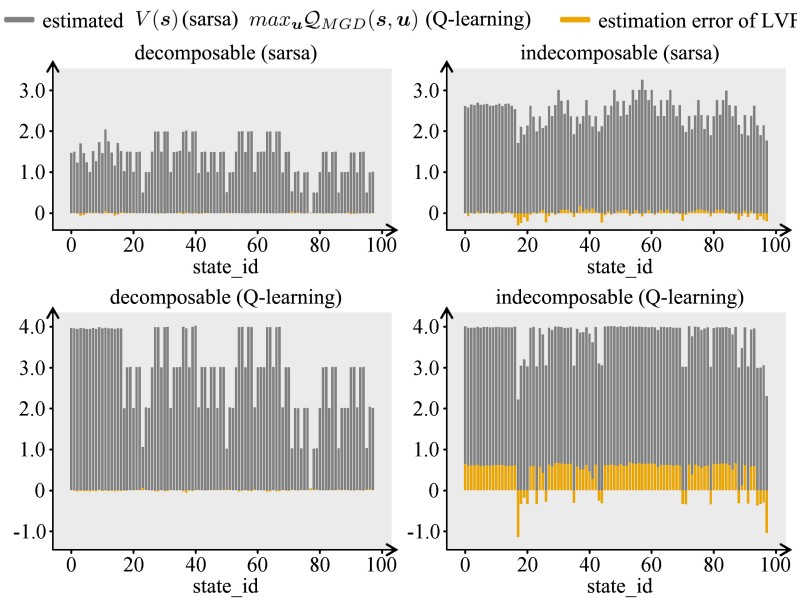

Figure 4: The estimation error of state value (sarsa) and max Q value (Q learning) in both decomposable and indecomposable Markov games.

From Fig.6 we can see that for $Q_{MGD}(\boldsymbol{s}, \boldsymbol{u})$ trained by sarsa value iteration, the difference between $V(\boldsymbol{s})$ and $V_{MGD}(\boldsymbol{s})$ is negligible in both decomposable and indecomposable Markov games. By contrast, for $Q_{MGD}(\boldsymbol{s}, \boldsymbol{u})$ trained by Q-learning value iteration, the difference between $max_{\boldsymbol{u}}Q(\boldsymbol{s}_t, \boldsymbol{u}_t)$ and $max_{\boldsymbol{u}}Q_{MGD}(\boldsymbol{s}_t, \boldsymbol{u}_t)$ is sizable in the indecomposable case. The experimental results indicate that although the true Q value function is not linearly factorizable in indecomposable Markov games, the representation target of a linearly factorized joint Q value function is still unbiased under the value iteration of sarsa manner. However, for a linearly factorized joint Q value function trained by Q-learning, the representation target is biased in indecomposable Markov games.

## 7 Conclusion

In this paper, we define the decomposability of Markov games and prove that the true Q value function is linearly factorizable if and only if the Markov game is decomposable. LVF is perfectly applicable in decomposable Markov games where each element of the MGD involves only a single agent. We also prove that in indecomposable Markov game, the estimate of state value function is still unbiased for LVF under the value iteration in sarsa manner. In addition to theoretical proofs, our conclusions are also verified in experiments on a toy game. To deal with the indecomposable Markov games, we explore the general form of value factorization functions that satisfy both IGM and CRC conditions. A common problem of these functions is the representation interference on the optimal true Q value function. To address this problem, we design a non-linear value factorization function that adaptively reweights the gradient contributed by different true Q values. Our method, namely QFRIS, is proved effective to address the representation interference in the experiments on matrix games. Besides, comparison with baselines in predator-prey and SMAC demonstrates the good convergence of our methods.

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

## A    PROOF OF PROPOSITION 1

### A.1    PROOF OF SUFFICIENCY

Given a decomposable Markov game $\mathcal{MG} =< \boldsymbol{S}, \boldsymbol{\mathcal{U}}, \mathcal{P}, \boldsymbol{r}, n, \gamma >$ and its decomposition $\{\mathcal{MG}_1, \mathcal{MG}_2, \cdots, \mathcal{MG}_k\}$, where $\mathcal{MG}_i =< \mathcal{S}_i, \hat{\mathcal{U}}_i, \mathcal{P}, r_i, n_i, \gamma >$. Firstly, consider the joint Q value function under the value iteration in sarsa manner. Suppose $\mathcal{Q}(\boldsymbol{s}_{t+1}, \boldsymbol{u}_{t+1})$ is linearly factorizable as $\mathcal{Q}(\boldsymbol{s}_{t+1}, \boldsymbol{u}_{t+1}) = \sum_{i=1}^{k} \mathcal{Q}_i(s_{i,t+1}, \hat{u}_{i,t+1})$. The state value function equals

$$V(\boldsymbol{s}_{t+1}) = \sum_{i=1}^{k} \int_{\boldsymbol{u}_{t+1}} \pi(\boldsymbol{u}_{t+1}|\boldsymbol{s}_{t+1}) \mathcal{Q}_i(s_{i,t+1}, \hat{u}_{i,t+1}) d\boldsymbol{u}_{t+1} \tag{14}$$

Let $\hat{u}_{\setminus i,t}$ denote the collection of the joint action except $\hat{u}_{i,t}$, i.e., $\hat{u}_{i,t} \cup \hat{u}_{\setminus i,t} = \boldsymbol{u}_t$ and $\hat{u}_{i,t} \cap \hat{u}_{\setminus i,t} = \emptyset$. Since the local policies are decentralized, the local actions are independent of each other. We have

$$V(\boldsymbol{s}_{t+1}) = \sum_{i=1}^{k} \int_{\boldsymbol{u}_{t+1}} \prod_{i=1}^{k} \pi_i(\hat{u}_{i,t+1}|s_{i,t+1}) \mathcal{Q}_i(s_{i,t+1}, \hat{u}_{i,t+1}) d\boldsymbol{u}_{t+1}$$

$$= \sum_{i=1}^{k} \int_{\hat{u}_{i,t+1}} \prod_{\hat{u}_{j,t+1}}^{\hat{u}_{\setminus i,t+1}} \int_{\hat{u}_{j,t+1}} \pi_j(\hat{u}_{j,t+1}|s_{j,t+1}) d\hat{u}_{j,t+1} \cdot \pi_i(\hat{u}_{i,t+1}|s_{i,t+1}) \mathcal{Q}_i(s_{i,t+1}, \hat{u}_{i,t+1}) d\hat{u}_{i,t+1}$$

$$= \sum_{i=1}^{k} \int_{\hat{u}_{i,t+1}} \pi_i(\hat{u}_{i,t+1}|s_{i,t+1}) \mathcal{Q}_i(s_{i,t+1}, \hat{u}_{i,t+1}) d\hat{u}_{i,t+1}$$

$$\tag{15}$$

Let $V_i(s_{i,t+1}) := \int_{\hat{u}_{i,t+1}} \pi_i(\hat{u}_{i,t+1}|s_{i,t+1}) \mathcal{Q}_i(s_{i,t+1}, \hat{u}_{i,t+1}) d\hat{u}_{i,t+1}$ denote the state value function of $\mathcal{MG}_i$. We have $V(\boldsymbol{s}_{t+1}) = \sum_{i=1}^{k} V_i(s_{i,t+1})$. According to Definition 1, the reward function is linearly factorizable in decomposable Markov game. The true Q value function equals

$$\mathcal{Q}(\boldsymbol{s}_t, \boldsymbol{u}_t) = \boldsymbol{r}(\boldsymbol{s}_t, \boldsymbol{u}_t) + \gamma \int_{\boldsymbol{s}_{t+1}} \mathcal{P}(\boldsymbol{s}_{t+1}|\boldsymbol{s}_t, \boldsymbol{u}_t) V(\boldsymbol{s}_{t+1}) d\boldsymbol{s}_{t+1}$$

$$= \sum_{i=1}^{k} \left( r_i(s_{i,t}, \hat{u}_{i,t}) + \gamma \int_{\boldsymbol{s}_{t+1}} \mathcal{P}(\boldsymbol{s}_{t+1}|\boldsymbol{s}_t, \boldsymbol{u}_t) V_i(s_{i,t+1}) d\boldsymbol{s}_{t+1} \right) \tag{16}$$

Note that $\mathcal{S}_i$ is a subspace of $\boldsymbol{S}$. We have

$$\int_{\boldsymbol{s}_{t+1}} \mathcal{P}(\boldsymbol{s}_{t+1}|\boldsymbol{s}_t, \boldsymbol{u}_t) V_i(s_{i,t+1}) d\boldsymbol{s}_{t+1} = \int_{s_{i,t+1}} \mathcal{P}(s_{i,t+1}|\boldsymbol{s}_t, \boldsymbol{u}_t) V_i(s_{i,t+1}) ds_{i,t+1} \tag{17}$$

According to the second condition in Definition 1, $P(s_{i,t+1}|s_{i,t}, \hat{u}_{i,t}) = P(s_{i,t+1}|\boldsymbol{s}_t, \boldsymbol{u}_t)$. We have

$$
\begin{aligned}
\mathcal{Q}(\boldsymbol{s}_t, \boldsymbol{u}_t) &= \sum_{i=1}^{k} \left( r_i(s_{i,t}, \hat{u}_{i,t}) + \gamma \int_{s_{i,t+1}} \mathcal{P}(s_{i,t+1}|\boldsymbol{s}_t, \boldsymbol{u}_t) V_i(s_{i,t+1}) ds_{i,t+1} \right) \\
&= \sum_{i=1}^{k} \left( r_i(s_{i,t}, \hat{u}_{i,t}) + \gamma \int_{s_{i,t+1}} \mathcal{P}(s_{i,t+1}|s_{i,t+1}, \hat{u}_{i,t+1}) V_i(s_{i,t+1}) ds_{i,t+1} \right) \quad (18) \\
&= \sum_{i=1}^{k} \mathcal{Q}_i(s_{i,t}, \hat{u}_{i,t})
\end{aligned}
$$

For the joint Q value function under the value iteration in $Q-$learning manner, suppose $\mathcal{Q}(\boldsymbol{s}_{t+1}, \boldsymbol{u}_{t+1})$ is linearly factorizable, we have

$$
\max_{\boldsymbol{u}_{t+1}} \mathcal{Q}(\boldsymbol{s}_{t+1}, \boldsymbol{u}_{t+1}) = \sum_{i=1}^{k} \max_{\hat{u}_{i,t+1}} \mathcal{Q}_i(s_{i,t+1}, \hat{u}_{i,t+1}) \quad (19)
$$

According to the properties of decomposable Markov game, we have

$$
\begin{aligned}
\mathcal{Q}(\boldsymbol{s}_t, \boldsymbol{u}_t) &= \boldsymbol{r}(\boldsymbol{s}_t, \boldsymbol{u}_t) + \gamma \int_{\boldsymbol{s}_{t+1}} \mathbb{P}(\boldsymbol{s}_{t+1}|\boldsymbol{s}_t, \boldsymbol{u}_t) \cdot \max_{\boldsymbol{u}_{t+1}} \mathcal{Q}(\boldsymbol{s}_{t+1}, \boldsymbol{u}_{t+1}) d\boldsymbol{s}_{t+1} \\
&= \sum_{i=1}^{k} \left( r_i(s_{i,t}, \hat{u}_{i,t}) + \gamma \int_{\boldsymbol{s}_{t+1}} \mathbb{P}(\boldsymbol{s}_{t+1}|\boldsymbol{s}_t, \boldsymbol{u}_t) \cdot \max_{\hat{u}_{i,t+1}} \mathcal{Q}_i(s_{i,t+1}, \hat{u}_{i,t+1}) d\boldsymbol{s}_{t+1} \right) \\
&= \sum_{i=1}^{k} \left( r_i(s_{i,t}, \hat{u}_{i,t}) + \gamma \int_{s_{i,t+1}} \mathbb{P}_i(s_{i,t+1}|s_{i,t}, \hat{u}_{i,t}) \max_{\hat{u}_{i,t+1}} \mathcal{Q}_i(s_{i,t+1}, \hat{u}_{i,t+1}) ds_{i,t+1} \right) \\
&= \sum_{i=1}^{k} \mathcal{Q}_i(s_{i,t}, \hat{u}_{i,t})
\end{aligned}
$$

$$(20)$$

We have proved that if $\mathcal{Q}(\boldsymbol{s}_{t+1}, \boldsymbol{u}_{t+1})$ is linearly factorizable, $\mathcal{Q}(\boldsymbol{s}_t, \boldsymbol{u}_t)$ is also linearly factorizable. For a finite Markov game, let $\forall i \in [1, k]$, $\mathcal{Q}(\boldsymbol{s}_T, \boldsymbol{u}_T) = \mathcal{Q}_i(s_{i,T}, \hat{u}_{i,T}) = 0$, where $T$ is the terminal time step. Since $\mathcal{Q}(\boldsymbol{s}_T, \boldsymbol{u}_T) = \sum_{i=1}^{k} \mathcal{Q}_i(s_{i,T}, \hat{u}_{i,T}) = 0$ is linearly factorizable, $\forall t \in [0, T]$, $\mathcal{Q}(\boldsymbol{s}_t, \boldsymbol{u}_t)$ is linearly factorizable. The factorizability of $\mathcal{Q}(\boldsymbol{s}_t, \boldsymbol{u}_t)$ in decomposable Markov games is proved.

## A.2 PROOF OF NECESSITY

Given a Markov game $\mathcal{MG} =< \boldsymbol{\mathcal{S}}, \boldsymbol{\mathcal{U}}, \mathcal{P}, r, n, \gamma >$, suppose the true Q value function $\mathcal{Q}(\boldsymbol{s}_t, \boldsymbol{u}_t)$ is linearly factorizable as $\mathcal{Q}(\boldsymbol{s}_t, \boldsymbol{u}_t) = \sum_{i=1}^{k} \mathcal{Q}_i(s_{i,t}, \hat{u}_{i,t})$, where $(\boldsymbol{s}_t, \boldsymbol{u}_t) \in \boldsymbol{\mathcal{S}} \times \boldsymbol{\mathcal{U}}$ and $(s_{i,t}, \hat{u}_{i,t}) \in \mathcal{S}_i \times \mathcal{U}_i$ ($i \in [1, k]$). $\mathcal{S}_i \times \mathcal{U}_i$ is a subspace of the joint state-action space (i.e., $\mathcal{S}_i \times \mathcal{U}_i \subset \boldsymbol{\mathcal{S}} \times \boldsymbol{\mathcal{U}}$). We have

$$
\mathcal{Q}(\boldsymbol{s}_t, \boldsymbol{u}_t) = \boldsymbol{r}(\boldsymbol{s}_t, \boldsymbol{u}_t) + \gamma \int_{\boldsymbol{s}_{t+1}} \mathcal{P}(\boldsymbol{s}_{t+1}|\boldsymbol{s}_t, \boldsymbol{u}_t) V(\boldsymbol{s}_{t+1}) d\boldsymbol{s}_{t+1} = \sum_{i=1}^{k} \mathcal{Q}_i(s_{i,t}, \hat{u}_{i,t}) \quad (21)
$$

Note that $r(\boldsymbol{s}_t, \boldsymbol{u}_t)$ is irrelevant to $V(\boldsymbol{s}_{t+1})$ because $r(\boldsymbol{s}_t, \boldsymbol{u}_t)$ is the reward of current time step but $\mathcal{Q}(\boldsymbol{s}_{t+1}, \boldsymbol{u}_{t+1})$ is determined by the policies, transitions and rewards of future time steps. Therefore, Eq.21 holds if and only if both $\boldsymbol{r}(\boldsymbol{s}_t, \boldsymbol{u}_t)$ and $\int_{\boldsymbol{s}_{t+1}} \mathcal{P}(\boldsymbol{s}_{t+1}|\boldsymbol{s}_t, \boldsymbol{u}_t) V(\boldsymbol{s}_{t+1}) d\boldsymbol{s}_{t+1}$ are linearly factorizable as

$$
\boldsymbol{r}(\boldsymbol{s}_t, \boldsymbol{u}_t) = \sum_{i=1}^{k} r_i(s_{i,t}, \hat{u}_{i,t})
$$

$$(22)$$

$$
\int_{\boldsymbol{s}_{t+1}} \mathcal{P}(\boldsymbol{s}_{t+1}|\boldsymbol{s}_t, \boldsymbol{u}_t) V(\boldsymbol{s}_{t+1}) d\boldsymbol{s}_{t+1} = \sum_{i=1}^{k} f_i(s_{i,t}, \hat{u}_{i,t})
$$

Let $s_{\backslash i,t}$ denote the other dimensions of $\boldsymbol{s}_t$ except $s_{i,t}$, i.e., $s_{i,t} \cup s_{\backslash i,t} = \boldsymbol{s}_t$ and $s_{i,t} \cap s_{\backslash i,t} = \emptyset$. Note that $\mathcal{Q}_i(s_{i,t}, \hat{u}_{i,t}) = r_i(s_{i,t}, \hat{u}_{i,t}) + \gamma \int_{s_{i,t+1}} \mathbb{P}_i(s_{i,t+1}|s_{i,t}, \hat{u}_{i,t}) V_i(s_{i,t+1}) ds_{i,t+1}$ and $\mathcal{Q}_i(s_{i,t}, \hat{u}_{i,t})$ is irrelevant to $(\boldsymbol{s}_{\backslash i,t}, \boldsymbol{u}_{\backslash i,t})$. We have

$$\mathbb{P}_i(s_{i,t+1}|s_{i,t}, \hat{u}_{i,t}) = \mathbb{P}_i(s_{i,t+1}|s_{i,t}, \boldsymbol{s}_{\backslash i,t}, \hat{u}_{i,t}, \boldsymbol{u}_{\backslash i,t}) = \mathbb{P}_i(s_{i,t+1}|, \boldsymbol{s}_t, \boldsymbol{u}_t) \tag{23}$$

According to Definition 1, $\mathcal{MG}$ is decomposable by $\{\mathcal{MG}_1, \mathcal{MG}_2, \cdots, \mathcal{MG}_k\}$.

## B  PROOF OF LEMMA 1

Given a Markov game $\mathcal{MG} = <\boldsymbol{S}, \boldsymbol{U}, \mathcal{P}, r, n, \gamma>$, $\{\mathcal{MG}_1, \mathcal{MG}_2, \cdots, \mathcal{MG}_k\}$ $(k \geq 2)$ is a decomposition of $\mathcal{MG}$. Suppose (1) $\mathcal{MG}_j' = <\mathcal{S}_j', \mathcal{U}_j', \mathcal{P}, r_j', n_j', \gamma>$ is decomposable by a non-empty subset of $\{\mathcal{MG}_1, \mathcal{MG}_2, \cdots, \mathcal{MG}_k\}$ for $\forall j \in [1, k_s]$; (2) $\cup_{j=1}^{k_s} \{\mathcal{S}_j' \times \hat{\mathcal{U}}_j'\} = \boldsymbol{S} \times \boldsymbol{U}$.

Let $A_j = [A_{1,j}, A_{2,j}, \cdots, A_{k,j}]$ $(j \in [1, k_s])$ denote an indicator vector, where $A_{i,j} = 1$ $(j \in [1, k])$ if $\mathcal{MG}_i$ is an element of the decomposition of $\mathcal{MG}_j'$, otherwise, $A_{i,j} = 0$. According to the definition of decomposable Markov game, $\mathcal{S}_i \times \hat{\mathcal{U}}_i \subset \mathcal{S}_j' \times \hat{\mathcal{U}}_j'$ if $A_{i,j} = 1$. We have $\hat{s}_j' = \cup_{i=1}^k A_{i,j} \cdot s_i$ and $\hat{u}_j' = \cup_{i=1}^k A_{i,j} \cdot \hat{u}_i$. Let $r_j'(s_j', \hat{u}_j')$ denote the reward function of $\mathcal{MG}_j'$, which is defined as

$$r_j'(s_j', \hat{u}_j') = \sum_{i=1}^k \frac{A_{i,j} \cdot r_i(s_i, \hat{u}_i)}{\sum_{j=1}^{k_s} A_{i,j}} \tag{24}$$

According to the second condition of Lemma 2, we have $\cup_{j=1}^{k_s} \hat{u}_j' = \boldsymbol{u}$. Note that $\cup_{i=1}^k \hat{u}_i = \boldsymbol{u}$ since $\{\mathcal{MG}_1, \mathcal{MG}_2, \cdots, \mathcal{MG}_k\}$ $(k \geq 2)$ is a decomposition of $\mathcal{MG}$. Therefore,

$$\begin{aligned}
\cup_{j=1}^{k_s} \hat{u}_j' &= \cup_{j=1}^{k_s} \left( \cup_{i=1}^k A_{i,j} \cdot \hat{u}_i \right) \\
&= \cup_{j=1}^{k_s} A_{i,j} \cdot \cup_{i=1}^k \hat{u}_i = \cup_{i=1}^k \hat{u}_i
\end{aligned} \tag{25}$$

which indicates $\forall j \in [1, k]$, $\sum_{j=1}^{k_s} A_{i,j} \geq 1$. In words, the denominator in Eq.24 are non-zero. The sum of the reward functions of $\{\mathcal{MG}_1', \mathcal{MG}_2', \cdots, \mathcal{MG}_{k_s}'\}$ equals

$$\begin{aligned}
\sum_{j=1}^{k_s} r_j'(s_j', \hat{u}_j') &= \sum_{j=1}^{k_s} \sum_{i=1}^k \frac{A_{i,j} \cdot r_i(s_i, \hat{u}_i)}{\sum_{j=1}^{k_s} A_{i,j}} = \sum_{i=1}^k \left( \sum_{j=1}^{k_s} \frac{A_{i,j} \cdot r_i(s_i, \hat{u}_i)}{\sum_{j=1}^{k_s} A_{i,j}} \right) \\
&= \sum_{i=1}^k \frac{\left( \sum_{j=1}^{k_s} A_{i,j} \right) \cdot r_i(s_i, \hat{u}_i)}{\sum_{j=1}^{k_s} A_{i,j}} = \sum_{i=1}^k r(s_i, \hat{u}_i) = \boldsymbol{r}(\boldsymbol{s}, \boldsymbol{u})
\end{aligned} \tag{26}$$

We have proved that the reward function is linearly factorizable on the collection of state-action spaces of $\{\mathcal{MG}_1', \mathcal{MG}_2', \cdots, \mathcal{MG}_{k_s}'\}$. Besides, since $\{\mathcal{MG}_1, \mathcal{MG}_2, \cdots, \mathcal{MG}_k\}$ is a decomposition of $\mathcal{MG}$, we have $\mathcal{P}(s_{i,t+1}|s_{i,t}, \hat{u}_{i,t}) = \mathcal{P}(s_{i,t+1}|\boldsymbol{s}_t, \boldsymbol{u}_t)$ for $\forall i \in [1, k]$. Note that $\hat{s}_j' = \cup_{i=1}^k A_{i,j} \cdot s_i$ and $\hat{u}_j' = \cup_{i=1}^k A_{i,j} \cdot \hat{u}_i$. We have

$$\begin{aligned}
\mathcal{P}(s_{j,t+1}'|s_{j,t}', \hat{u}_{j,t}') &= \mathcal{P}(\cup_{i=1}^k A_{i,j} \cdot s_{i,t+1}| \cup_{i=1}^k A_{i,j} \cdot s_{i,t}, \cup_{i=1}^k A_{i,j} \cdot \hat{u}_{i,t}) \\
&= \mathcal{P}(\cup_{i=1}^k A_{i,j} \cdot s_{i,t+1}|\boldsymbol{s}_t, \boldsymbol{u}_t) = \mathcal{P}(s_{j,t+1}'|\boldsymbol{s}_t, \boldsymbol{u}_t)
\end{aligned} \tag{27}$$

According to the definition of decomposable Markov game, $\{\mathcal{MG}_1', \mathcal{MG}_2', \cdots, \mathcal{MG}_{k_s}'\}$ is a decomposition of $\mathcal{MG}$.

## C  RANK OF THE COEFFICIENT MATRIX

In indecomposable Markov games with discrete action spaces, the representation of true Q value function is equivalent to solving the linear equation system

$$\left\{ \mathcal{Q}(\boldsymbol{s}, \boldsymbol{u}) = \sum_{i=1}^n w_i Q_i(\boldsymbol{s}, u_i) + V(\boldsymbol{s}) \right\}_{\forall \boldsymbol{u} \in \boldsymbol{U}} \tag{28}$$

For simplicity, we omit the state value function $V(s)$. Let $\{1, 2, \cdots, m\}$ denote the discrete local action space. The joint Q value function can be written as

$$
\begin{aligned}
Q(s, u) &= \mathbb{I}(u_1 = 1)Q_1(1) + \mathbb{I}(u_1 = 2)Q_1(2) + \cdots + \mathbb{I}(u_1 = m)Q_1(m) \\
&+ \mathbb{I}(u_2 = 1)Q_2(1) + \mathbb{I}(u_2 = 2)Q_2(2) + \cdots + \mathbb{I}(u_2 = m)Q_2(m) \\
&+ \cdots \\
&+ \mathbb{I}(u_n = 1)Q_n(1) + \mathbb{I}(u_n = 2)Q_n(2) + \cdots + \mathbb{I}(u_n = m)Q_n(m) \\
&= \left[ \overbrace{\mathbb{I}(u_1 = 1) \ \cdots \ \mathbb{I}(u_1 = m)}^{agent\ 1} \ \overbrace{\mathbb{I}(u_2 = 1) \ \cdots \ \mathbb{I}(u_2 = m)}^{agent\ 2} \ \cdots \ \overbrace{\mathbb{I}(u_n = 1) \ \cdots \ \mathbb{I}(u_n = m)}^{agent\ n} \right] \\
&\quad \cdot \left[ \overbrace{Q_1(1) \ \cdots \ Q_1(m)}^{agent\ 1} \ \overbrace{Q_2(1) \ \cdots \ Q_2(m)}^{agent\ 2} \ \cdots \ \overbrace{Q_n(1) \ \cdots \ Q_n(m)}^{agent\ n} \right]^\top
\end{aligned}
$$
(29)

Here we omit the states in all inputs. In Markov game with discrete action space, the true Q values of the joint actions' all permutations constitute the complete set of the representation targets. For example, the all permutations of 2-agent joint actions are

$$
\left[ \overbrace{(1,1) \ (2,1) \ \cdots \ (m,1)}^{(\cdot,1)} \ \overbrace{(1,2) \ (2,2) \ \cdots \ (m,2)}^{(\cdot,2)} \ \cdots \ \cdots \ \overbrace{(1,m) \ (2,m) \ \cdots \ (m,m)}^{(\cdot,m)} \right]^\top
$$
(30)

Eq.28 is equivalent to the following matrix equation in 2-agent cases:

$$
\mathcal{A}^2 \times \vec{Q}_{loc}^2 = \vec{\mathcal{Q}}^2
$$
(31)

where

$$
\mathcal{A}^2 = \begin{bmatrix}
\overbrace{\begin{matrix} 1 & 0 & \cdots & 0 \end{matrix}}^{agent\ 1} & \overbrace{\begin{matrix} 1 & 0 & \cdots & 0 \end{matrix}}^{agent\ 2} \\
0 & 1 & \cdots & 0 & 1 & 0 & \cdots & 0 \\
\vdots & \vdots & \ddots & \vdots & \vdots & \vdots & \ddots & \vdots \\
0 & 0 & \cdots & 1 & 1 & 0 & \cdots & 0 \\
& \vdots & & & & \vdots & & \\
1 & 0 & \cdots & 0 & 0 & 0 & \cdots & 1 \\
0 & 1 & \cdots & 0 & 0 & 0 & \cdots & 1 \\
\vdots & \vdots & \ddots & \vdots & \vdots & \vdots & \ddots & \vdots \\
0 & 0 & \cdots & 1 & 0 & 0 & \cdots & 1
\end{bmatrix}, \quad
\vec{Q}_{loc}^2 = \begin{bmatrix} Q_1(1) \\ Q_1(2) \\ \vdots \\ Q_1(m) \\ Q_2(1) \\ Q_2(2) \\ \vdots \\ Q_2(m) \end{bmatrix}, \quad
\vec{\mathcal{Q}}^2 = \begin{bmatrix} \mathcal{Q}(1,1) \\ \mathcal{Q}(2,1) \\ \vdots \\ \mathcal{Q}(m,1) \\ \vdots \\ \vdots \\ \mathcal{Q}(1,m) \\ \mathcal{Q}(2,m) \\ \vdots \\ \mathcal{Q}(m,m) \end{bmatrix}
$$
(32)

The coefficient matrix $\mathcal{A}^2$ can be represented by

$$
\mathcal{A}^2 = \begin{bmatrix} E_m & A_1^2 \\ E_m & A_2^2 \\ \vdots & \vdots \\ E_m & A_m^2 \end{bmatrix}, \quad A_i^2 = \begin{bmatrix} O_{i-}^2 & \vec{I}^2 & O_{i+}^2 \end{bmatrix}
$$
(33)

where $E_m$ is an $m$-dimensional unit matrix. $O_{i-}^2$ and $O_{i+}^2$ are zero matrices of size $m \times i$ and $m \times (m - i - 1)$ ($i \in [0, m - 1]$), respectively. $\vec{I}^2$ is an $m$-dimensional column vector with all 1 elements. Note that $rk\left( \begin{bmatrix} E_m & A_1^2 \\ E_m & A_2^2 \end{bmatrix} \right) = m + 1$. We have $rk(\mathcal{A}^2) = m + (m - 1) = 2m - 1$. Now we extend the 2-agent case to the 3-agent, where

$$
\mathcal{A}^3 = \begin{bmatrix} \mathcal{A}^2 & A_1^3 \\ \mathcal{A}^2 & A_2^3 \\ \vdots & \vdots \\ \mathcal{A}^2 & A_m^3 \end{bmatrix}, \quad A_i^3 = \begin{bmatrix} O_{i-}^3 & \vec{I}^3 & O_{i+}^3 \end{bmatrix}
$$
(34)

$O_{i-}^3$ and $O_{i+}^3$ are zero matrices of size $m^2 \times i$ and $m^2 \times (m-i-1)$ ($i \in [0, m-1]$), respectively. $\vec{I}^3$ is an $m^2$-dimensional column vector with all 1 elements. We have $rk(\mathcal{A}^3) = rk(\mathcal{A}^2) + m - 1 = 3m - 2$. For the $n$-agent case, we can infer that

$$rk(\mathcal{A}^n) = rk(\mathcal{A}^{n-1}) + m - 1 = rk(\mathcal{A}^2) + (n-2) \cdot (m-1) = n(m-1) + 1 \tag{35}$$

## D  PROOF OF PROPOSITION 2

Eq.28 is equivalent to the following matrix equation

$$\mathcal{A}^n \times \vec{Q}_{loc}^n = \vec{\mathcal{Q}}^n \tag{36}$$

The expression of $\mathcal{A}^n$, $\vec{Q}_{loc}^n$ and $\vec{\mathcal{Q}}^n$ can be inferred from Eq.32. We consider the worst case where the augment matrix is full rank, i.e., $rk([\mathcal{A}^n \quad \vec{\mathcal{Q}}^n]) = m^n$. Note that $m^n > n(m-1) + 1$ for $\forall m, n \in [2, \infty)$. The matrix equation is overdetermined, which can be solved by least square method. Let $\vec{\pi}^n$ denote the vector of all permutations of the joint action's probabilities. Notice $\sqrt{\vec{\pi}^n} \cdot (\mathcal{A}^n \times \vec{Q}_{loc}^n) = (\sqrt{\vec{\pi}^n} \cdot \mathcal{A}^n) \times \vec{Q}_{loc}^n$. The aim of least square method is:

$$min \ \vec{\pi}^n \cdot ||\mathcal{A}^n \times \vec{Q}_{loc}^n - \vec{\mathcal{Q}}^n|| = min \ ||(\sqrt{\vec{\pi}^n} \cdot \mathcal{A}^n) \times \vec{Q}_{loc}^n - \sqrt{\vec{\pi}^n} \cdot \vec{\mathcal{Q}}^n|| \tag{37}$$

$\vec{Q}_{loc}^{n*}$ is the least square solution if and only if the following holds:

$$(\sqrt{\vec{\pi}^n} \cdot \mathcal{A}^n)^\top \times (\sqrt{\vec{\pi}^n} \cdot \mathcal{A}^n) \times \vec{Q}_{loc}^{n*} = (\sqrt{\vec{\pi}^n} \cdot \mathcal{A}^n)^\top \times (\sqrt{\vec{\pi}^n} \cdot \vec{\mathcal{Q}}^n) \tag{38}$$

Let $\vec{Q}_{jt}^{n*}$ denote the vector of all permutations of the joint Q values under the least square solution. Notice that $\vec{Q}_{jt}^{n*} = \mathcal{A}^n \times \vec{Q}_{loc}^{n*}$. We have

$$(\sqrt{\vec{\pi}^n} \cdot \mathcal{A}^n)^\top \times (\sqrt{\vec{\pi}^n} \cdot \mathcal{A}^n) \times \vec{Q}_{loc}^{n*} = (\sqrt{\vec{\pi}^n} \cdot \mathcal{A}^n)^\top \times (\sqrt{\vec{\pi}^n} \cdot \vec{Q}_{jt}^n) \tag{39}$$

Combining Eq.38 with Eq.39, we have

$$\begin{aligned}
(\sqrt{\vec{\pi}^n} \cdot \mathcal{A}^n)^\top \times (\sqrt{\vec{\pi}^n} \cdot \vec{\mathcal{Q}}^n) &= \mathcal{A}^{n\top} \times (\vec{\pi}^n \cdot \vec{\mathcal{Q}}^n) \\
&= (\sqrt{\vec{\pi}^n} \cdot \mathcal{A}^n)^\top \times (\sqrt{\vec{\pi}^n} \cdot \vec{Q}_{jt}^n) = \mathcal{A}^{n\top} \times (\vec{\pi}^n \cdot \vec{Q}_{jt}^n)
\end{aligned} \tag{40}$$

According to Eq.34, we have

$$\mathcal{A}^{n\top} = \begin{bmatrix} \mathcal{A}^{n-1\top} & \mathcal{A}^{n-1\top} & \cdots & \mathcal{A}^{n-1\top} \\ A_1^{n\top} & A_2^{n\top} & \cdots & A_m^{n\top} \end{bmatrix}, \quad A_i^{n\top} = \begin{bmatrix} O_{i-}^{n\top} \\ \vec{I}^{n\top} \\ O_{i+}^{n\top} \end{bmatrix} \tag{41}$$

where $O_{i-}^{n\top}$ and $O_{i+}^{n\top}$ are zero matrices of size $i \times m^{n-1}$ and $(m-i-1) \times m^{n-1}$ ($i \in [0, m-1]$), respectively. $\vec{I}^{n\top}$ is an $m^{n-1}$-dimensional row vector with all 1 elements. Referring to Eq.40 and Eq.41, we have

$$\begin{bmatrix} A_1^{n\top} & A_2^{n\top} & \cdots & A_m^{n\top} \end{bmatrix} \times (\vec{\pi}^n \cdot \vec{Q}_{jt}^n) = \begin{bmatrix} A_1^{n\top} & A_2^{n\top} & \cdots & A_m^{n\top} \end{bmatrix} \times (\vec{\pi}^n \cdot \vec{\mathcal{Q}}^n) \tag{42}$$

In words, $\forall i \in [1, m]$, the following holds

$$\sum_{\boldsymbol{u}_{\backslash 1}}^{m^{n-1}} \pi(u_1 = i, \boldsymbol{u}_{\backslash 1}|\boldsymbol{s})Q(\boldsymbol{s}, u_1 = i, \boldsymbol{u}_{\backslash 1}) = \sum_{\boldsymbol{u}_{\backslash 1}}^{m^{n-1}} \pi(u_1 = i, \boldsymbol{u}_{\backslash 1}|\boldsymbol{s})\mathcal{Q}(\boldsymbol{s}, u_1 = i, \boldsymbol{u}_{\backslash 1}) \tag{43}$$

where $\boldsymbol{u}_{\backslash 1}$ denotes the group of all actions except $u_1$. Summing up the equations from $u_1 = 1$ to $u_1 = m$, we have

$$\begin{aligned}
\sum_{i=1}^{m} \sum_{\boldsymbol{u}_{\backslash 1}}^{m^{n-1}} \pi(u_1 = i, \boldsymbol{u}_{\backslash 1}|\boldsymbol{s})Q(\boldsymbol{s}, u_1 = i, \boldsymbol{u}_{\backslash 1}) &= \sum_{\boldsymbol{u}}^{m^n} \pi(\boldsymbol{u}|\boldsymbol{s})Q(\boldsymbol{s}, \boldsymbol{u}) \\
= \sum_{i=1}^{m} \sum_{\boldsymbol{u}_{\backslash 1}}^{m^{n-1}} \pi(u_1 = i, \boldsymbol{u}_{\backslash 1}|\boldsymbol{s})\mathcal{Q}(\boldsymbol{s}, u_1 = i, \boldsymbol{u}_{\backslash 1}) &= \sum_{\boldsymbol{u}}^{m^n} \pi(\boldsymbol{u}|\boldsymbol{s})\mathcal{Q}(\boldsymbol{s}, \boldsymbol{u})
\end{aligned} \tag{44}$$

$\sum_{\boldsymbol{u}}^{m^n} \pi(\boldsymbol{u}|\boldsymbol{s})Q(\boldsymbol{s}, \boldsymbol{u})$ is the state value estimated by the joint Q value function of LVF and $\sum_{\boldsymbol{u}}^{m^n} \pi(\boldsymbol{u}|\boldsymbol{s})\mathcal{Q}(\boldsymbol{s}, \boldsymbol{u})$ is the actual state value. In words, the estimate of the state value function is unbiased for LVF under the value iteration in sarsa manner.

# E    EXAMPLES OF EXTENDED LVF AND NVF

## E.1    EXAMPLES OF EXTENDED LVF

An example of linear value factorization function is QTRAN (Son et al., 2019). Let $\mathcal{M}_{set}(\boldsymbol{u}) = \mathcal{M}_1(\boldsymbol{u})$ and $C_{Q,i} = 1$ ($i \in [1, n]$). We have

$$Q(\boldsymbol{u}) = \mathcal{F}(Q_{set}(\boldsymbol{u}), \mathcal{M}_1(\boldsymbol{u})) = \sum_{i=1}^{n} Q_i(u_i) + \mathcal{M}_1(\boldsymbol{u}) + C_{\mathcal{F}} \tag{45}$$

In QTRAN, $Q(\boldsymbol{u})$ and $C_{\mathcal{F}}$ refer to the joint Q value function (i.e., $Q_{jt}$) and the state value function (i.e., $V_{jt}$), respectively. $\mathcal{M}_1(\boldsymbol{u})$ is the error between $Q(\boldsymbol{u})$ and $\sum_{i=1}^{n} Q_i(u_i) + C_{\mathcal{F}}$. To ensure the IGM principle, QTRAN applies two regularizations to regulate that $\mathcal{M}_1(\boldsymbol{u}) \leq \mathcal{M}_1(\boldsymbol{u}_{gre}) = 0$.

To be more compact, let $\mathcal{M}_{set}(\boldsymbol{u}) = \mathcal{M}_1(\boldsymbol{u})$, $C_{Q,i} = 1$ ($i \in [1, n]$) and $C_{\mathcal{F}} = -\mathcal{M}_1(\boldsymbol{u}_{gre})$. We have

$$Q(\boldsymbol{u}) = \mathcal{F}(Q_{set}(\boldsymbol{u}), \mathcal{M}_1(\boldsymbol{u})) = \sum_{i=1}^{n} Q_i(u_i) - \mathcal{M}^d(\boldsymbol{u}) \tag{46}$$

where $\mathcal{M}^d(\boldsymbol{u}) = -\mathcal{M}_1(\boldsymbol{u}) + \mathcal{M}_1(\boldsymbol{u}_{gre})$. According to Condition 1.2, we have $\mathcal{M}^d : \mathcal{U} \to [0, \infty)$, $\mathcal{M}^d(\boldsymbol{u}) = 0$ if $\boldsymbol{u} = \boldsymbol{u}_{gre}$ else $\mathcal{M}^d(\boldsymbol{u}) \geq 0$. $\mathcal{M}^d(\boldsymbol{u})$ can be modelled by distance or indicator functions. Examples of $\mathcal{M}^d(\boldsymbol{u})$ are listed as follows:

$$\begin{aligned} \mathcal{M}_1^d(\boldsymbol{u}) &= |\mathcal{G}(\boldsymbol{u}) - \mathcal{G}(\boldsymbol{u}_{gre})| \\ \mathcal{M}_2^d(\boldsymbol{u}) &= [\mathcal{G}(\boldsymbol{u}) - \mathcal{G}(\boldsymbol{u}_{gre})]^2 \\ \mathcal{M}_3^d(\boldsymbol{u}) &= I(\boldsymbol{u} = \boldsymbol{u}_{gre})e^{\mathcal{G}(\boldsymbol{u})} \end{aligned} \tag{47}$$

where $\mathcal{G} : \mathcal{U} \to \mathbb{R}$. $I(\boldsymbol{u} = \boldsymbol{u}_{gre})$ is an indicator function and $I(\boldsymbol{u} = \boldsymbol{u}_{gre}) = 1$ if $\boldsymbol{u} = \boldsymbol{u}_{gre}$ else $I(\boldsymbol{u} = \boldsymbol{u}_{gre}) = 0$.

## E.2    EXAMPLES OF NVF FUNCTIONS

Consider parameterizing the operator directly. The IGM principle requires $\mathcal{F}_\theta(Q_{set}(\boldsymbol{\tau}, \boldsymbol{u})) \leq \mathcal{F}_\theta(Q_{set}(\boldsymbol{\tau}, \boldsymbol{u}))$. Let

$$\mathcal{F}_\theta(Q_{set}(\boldsymbol{\tau}, \boldsymbol{u})) = -\mathcal{F}_\theta^d(Q_{set}(\boldsymbol{\tau}, \boldsymbol{u})) + \mathcal{F}_\theta^c(Q_{set}(\boldsymbol{\tau}, \boldsymbol{u}_{gre})) \tag{48}$$

where $\mathcal{F}_\theta^d : \mathcal{U} \to [0, \infty)$, $\mathcal{F}_\theta^d(Q_{set}(\boldsymbol{u})) = 0$ if $\boldsymbol{u} = \boldsymbol{u}_{gre}$ else $\mathcal{F}_\theta^d(Q_{set}(\boldsymbol{u})) \geq 0$, which can be modelled by Eq.47. $\mathcal{F}_\theta^c(Q_{set}(\boldsymbol{u}_{gre})$ is trained to model $\mathcal{Q}_{\boldsymbol{u}_{gre}}$. $\mathcal{F}_\theta(Q_{set}(\boldsymbol{u}))$ is directly optimized by value iteration, by which $Q_i(u_i)$ is updated. The final objective requires both $max\ Q_i(u_i) = Q_i(u_i^*)$ and $max\ \mathcal{F}_\theta(Q_{set}(\boldsymbol{u})) = \mathcal{F}_\theta(Q_{set}(\boldsymbol{u}^*))$. Let $\mathcal{F}_{\theta,i}'(\boldsymbol{u})$ denote the partial derivatives for $Q_i(u_i)$ ($i \in [1, n]$), i.e., $\mathcal{F}_{\theta,i}'(\boldsymbol{u}) = \partial \mathcal{F}_\theta(Q_{set}(\boldsymbol{u}))/\partial Q_i(u_i)$. As $\mathcal{F}_\theta(Q_{set}(\boldsymbol{u}))$ is typically modelled by neural networks, there is a risk that $\mathcal{F}_{\theta,i}'(\boldsymbol{u}) < 0$. A negative $\mathcal{F}_{\theta,i}'(\boldsymbol{u})$ means the increase in $\mathcal{F}_\theta(Q_{set}(\boldsymbol{u}))$ result in the decrease of $Q_i(u_i)$, which misleads the update of $Q_i(u_i)$. For linear value factorization function, the problem can be avoided by setting $w_i > 0$ since $\mathcal{F}_{\theta,i}'(\boldsymbol{u}) = w_i$.

Consider introducing parameterized modules. For brevity, we only consider the model $\mathcal{M}_j(\hat{u}_j)$ conditioned on the joint action spaces, i.e., $\mathcal{U}_j = \mathcal{U}$. Refer to Eq.48 and let $\mathcal{F}(Q_{set}(\boldsymbol{u}), \mathcal{M}_{set}(\boldsymbol{u})) = -\mathcal{F}^d(Q_{set}(\boldsymbol{u}), \mathcal{M}_{set}(\boldsymbol{u})) + \mathcal{F}^c(Q_{set}(\boldsymbol{u}_{gre}), \mathcal{M}_{set}(\boldsymbol{u}_{gre}))$. We require $\mathcal{F}^d : \mathcal{U} \to [0, \infty)$ and $\mathcal{F}^d(Q_{set}(\boldsymbol{u}), \mathcal{M}_{set}(\boldsymbol{u})) = 0$ if $\boldsymbol{u} = \boldsymbol{u}_{gre}$ else $\mathcal{F}^d(Q_{set}(\boldsymbol{u}), \mathcal{M}_{set}(\boldsymbol{u})) \geq 0$ (# Condition 2.2). $\mathcal{F}^c(Q_{set}(\boldsymbol{u}_{gre})$ is trained to model $\mathcal{Q}_{\boldsymbol{u}_{gre}}$.

Examples of $\mathcal{F}^d(Q_{set}(\boldsymbol{u}), \mathcal{M}_{set}(\boldsymbol{u}))$ are listed as follows:

$$\mathcal{F}_1^d(Q_{set}(\boldsymbol{u}), \mathcal{M}_{set}(\boldsymbol{u})) = |\mathcal{M}_1(\boldsymbol{u})| \cdot \sum_{i=1}^{n} [Q_i(u_{i,gre}) - Q_i(u_i)]$$

$$\mathcal{F}_2^d(Q_{set}(\boldsymbol{u}), \mathcal{M}_{set}(\boldsymbol{u})) = |\mathcal{M}_1(\boldsymbol{u}) - \mathcal{M}_1(\boldsymbol{u}_{gre})| \cdot \prod_{i=1}^{n} e^{-Q_i(u_i)} \tag{49}$$

$$\mathcal{F}_3^d(Q_{set}(\boldsymbol{u}), \mathcal{M}_{set}(\boldsymbol{u})) = I(\boldsymbol{u} = \boldsymbol{u}_{gre})e^{\mathcal{M}_1(\boldsymbol{u})} + \sum_{i=1}^{n} [Q_i(u_{i,gre}) - Q_i(u_i)]^2$$

A variant of $\mathcal{F}_1^d(Q_{set}(\boldsymbol{u}), \mathcal{M}_{set}(\boldsymbol{u}))$ is $\sum_{i=1}^n |\mathcal{M}_1(\boldsymbol{u})| \cdot [Q_i(u_{i,gre}) - Q_i(u_i)]$. Let $\mathcal{F}(Q_{set}(\boldsymbol{u}_{gre}), \mathcal{M}_{set}(\boldsymbol{u}_{gre})) = \sum_{i=1}^n Q_i(u_{i,gre})$. The joint Q value function equals

$$Q(\boldsymbol{u}) = \mathcal{F}(Q_{set}(\boldsymbol{u}), \mathcal{M}_{set}(\boldsymbol{u})) = -\sum_{i=1}^n |\mathcal{M}_1(\boldsymbol{u})| \cdot [Q_i(u_{i,gre}) - Q_i(u_i)] + \sum_{i=1}^n Q_i(u_{i,gre}) \quad (50)$$

which is exact the joint Q value function of QPLEX (Wang et al., 2020b).

# F RELATED WORKS

Independent learning has been introduced in fully cooperative multi-agent tasks for a long time (Tan, 1993a). In tasks with small number of agents, independent proximal policy optimization (PPO) with agent-specific reward functions is able to acquire strategies on the level of human experts. For better scalability, recent works turn to automatic credit assignment under reward functions shared by the team. Meanwhile, by introducing global information in the training of local policies, centralized training with decentralized execution (CTDE) achieve great success in complex cooperative MARL tasks. As a simple and effective approach to achieve credit assignment in the paradigm of CTDE, value decomposition recently gains wide attention.

## F.1 LINEAR VALUE FACTORIZATION

There is a series of implementations of linear value factorization. VDN (Sunehag et al., 2017) obtains the joint Q value function by simply adding all local Q value functions together and update the joint Q value function by $Q-$learning value iteration. Based on VDN, QMIX (Rashid et al., 2018) extracts a set of weights form the global state and applies them to the local Q value functions. SMIX (Wen et al., 2020) and Qatten (Yang et al., 2020) share the same value factorization function with QMIX. SMIX replaces the TD(0) $Q-$learning target with a TD($\lambda$) sarsa target. Qatten introduces an attention network before the mixing network. All methods above suffer form relative overgeneralization due to the representation limitation of the joint Q value function, i.e., there would be multiple possible convergence.

## F.2 VALUE FACTORIZATION FOR INDECOMPOSABLE MARKOV GAMES

There are various works to address relative overgeneralization, which can be summarized into two categories. The first is biased representation. The basic idea is reducing the representation errors of the $Q-$learning targets at the expense of increased representation errors of non-maximal joint Q values. WQMIX (Rashid et al., 2020) alleviates the estimate error of the $Q-$learning targets by attaching more weights on the representation of the joint Q values for potential optimal actions. In practice, a weight $\alpha \in (0, 1)$ is applied to the samples with lower targets than expected. The $Q-$learning targets are unbiased when $\alpha = 0$. GVR (Wan et al., 2021) achieves approximatively unbiased estimate of the $Q-$learning targets by target shaping. The former reshapes the targets of joint Q values lower than expected, while the latter reshapes the targets matrices into a monotonic from. Biased representation alleviates but does not eliminate the representation errors. Besides, these methods rely on the identification of potential optimal actions, which would introduce extra errors in the training. Another route to address the indecomposable Markov game is completing the representation capability of the joint Q value function under the IGM principle, i.e., introducing value factorization functions with both IGM and CRC properties, e.g., Qtran and QPLEX.

Qtran (Son et al., 2019) adopts a linear value factorization function as $Q(\boldsymbol{s}, \boldsymbol{u}) = \sum_{i=1}^n Q_i(\tau_i, u_i) + \mathcal{M}_1(\boldsymbol{s}, \boldsymbol{u}) + V(\boldsymbol{s})$ (Eq.45). The IGM principle requires $\mathcal{M}_1(\boldsymbol{s}, \boldsymbol{u}) \leq \mathcal{M}_1(\boldsymbol{s}, \boldsymbol{u}_{gre}) = 0$. However, instead of modelling $\mathcal{M}_1(\boldsymbol{s}, \boldsymbol{u})$ explicitly, Qtran models $Q(\boldsymbol{s}, \boldsymbol{u})$ and represent $\mathcal{M}_1(\boldsymbol{s}, \boldsymbol{u})$ by $\mathcal{M}_1(\boldsymbol{s}, \boldsymbol{u}) = Q(\boldsymbol{s}, \boldsymbol{u}) - \sum_{i=1}^n Q_i(\tau_i, u_i) - V(\boldsymbol{s})$. As a result, (1) Qtran does not satisfy the IGM principle strictly. To approximate the IGM principle, Qtran applies a multi-stage training to regulate that $\mathcal{M}_1(\boldsymbol{s}, \boldsymbol{u}) \leq \mathcal{M}_1(\boldsymbol{s}, \boldsymbol{u}_{gre}) = 0$. In the first stage, $Q(\boldsymbol{s}, \boldsymbol{u})$ is trained to approximate the true Q value function; In the second stage, $V(\boldsymbol{s})$ is trained to meet $\mathcal{M}_1(\boldsymbol{u}_{gre}) = 0$, i.e., $V(\boldsymbol{s}) = Q_{jt}(\boldsymbol{s}, \boldsymbol{u}_{gre}) - \sum_{i=1}^n Q_i(\tau_i, u_{i,gre})$; In the third stage, the local value functions are trained to meet $\mathcal{M}_1(\boldsymbol{s}, \boldsymbol{u}) \leq 0$, where the local policies are updated only if $\mathcal{M}_1(\boldsymbol{s}, \boldsymbol{u}) > 0$, i.e., $Q(\boldsymbol{s}, \boldsymbol{u}) - \sum_{i=1}^n Q_i(\tau_i, u_i) > V(\boldsymbol{s})$. As a result, (2) the estimate errors of $Q(\boldsymbol{s}, \boldsymbol{u})$ and $V(\boldsymbol{s})$ magnify

the estimate errors of local Q value functions. A simple method to overcome the above two defects of Qtran is modelling $\mathcal{M}_1(\boldsymbol{s}, \boldsymbol{u})$ explicitly with functions that satisfies $\mathcal{M}_1(\boldsymbol{s}, \boldsymbol{u}) \leq \mathcal{M}_1(\boldsymbol{s}, \boldsymbol{u}_{gre}) = 0$, e.g., distance functions. Then represent $Q(\boldsymbol{s}, \boldsymbol{u})$ by $\mathcal{M}_1(\boldsymbol{s}, \boldsymbol{u})$ and train $Q(\boldsymbol{s}, \boldsymbol{u})$ end-to-end.

QPLEX (Wang et al., 2020b) adopts a non-linear value factorization function as $Q(\boldsymbol{s}, \boldsymbol{u}) = -\sum_{i=1}^{n} |\mathcal{M}_i(\boldsymbol{s}, \boldsymbol{u})| \cdot [Q_i(\tau_i, u_{i,gre}) - Q_i(\tau_i, u_i)] + \sum_{i=1}^{n} Q_i(\tau_i, u_{i,gre})$ (Eq.50). A set of models $\{\mathcal{M}_1, \mathcal{M}_2, \cdots, \mathcal{M}_n\}$ is introduced to constitute $Q(\boldsymbol{s}, \boldsymbol{u})$ and $Q(\boldsymbol{s}, \boldsymbol{u})$ is trained end-to-end. However, QPLEX is easily trapped in local optimums for the following two reasons. (1) Suppose current greedy action is not the optimal action, i.e., $\boldsymbol{u}_{gre} \neq \boldsymbol{u}^*$. The convergence to the global optimum requires $Q(\boldsymbol{s}, \boldsymbol{u}^*) > Q(\boldsymbol{s}, \boldsymbol{u}_{gre})$. Notice $\partial Q(\boldsymbol{s}, \boldsymbol{u}^*) / \partial Q_i(\tau_i, u_i^*) = |\mathcal{M}_i(\boldsymbol{s}, \boldsymbol{u}^*)|$ ($i \in \{1, n\}$), which can be viewed as a weight of the sample. As $Q(\boldsymbol{s}, \boldsymbol{u}^*)$ increases, $|\mathcal{M}_i(\boldsymbol{s}, \boldsymbol{u}^*)|$, i.e., the weight of the optimal sample decreases to 0. (2) For $\boldsymbol{u}' = \{u_1^*, u_{2,gre}, u_{3,gre}, \cdots, u_{n,gre}\}$, we have $Q(\boldsymbol{s}, \boldsymbol{u}') = -|\mathcal{M}_1(\boldsymbol{s}, \boldsymbol{u}')| \cdot [Q_1(\tau_1, u_{1,gre}) - Q_1(\tau_1, u_1^*)] + Q(\boldsymbol{s}, \boldsymbol{u}_{gre})$, where $Q(\boldsymbol{s}, \boldsymbol{u}_{gre}) = \sum_{i=1}^{n} Q_i(\tau_i, u_{i,gre})$. $-|\mathcal{M}_1(\boldsymbol{s}, \boldsymbol{u}')| \cdot [Q_1(\tau_1, u_{1,gre}) - Q_1(\tau_1, u_1^*)]$ is trained to approximate $\mathcal{Q}(\boldsymbol{s}, \boldsymbol{u}') - \mathcal{Q}(\boldsymbol{s}, \boldsymbol{u}_{gre})$. The convergence to the global optimum requires $Q_i(\tau_1, u_1^*) > Q_i(\tau_1, u_{1,gre})$. As $Q_1(\tau_1, u_1^*)$ increases and approximates $Q_1(\tau_1, u_{1,gre})$, $|\mathcal{M}_1(\boldsymbol{s}, \boldsymbol{u}')|$, i.e., the weight of the sample $\mathcal{Q}(\boldsymbol{s}, \boldsymbol{u}')$ increases sharply. $Q_1(\tau_1, u_1^*)$ is updated by all samples involving $u_1^*$, e.g., $\mathcal{Q}(\boldsymbol{s}, \boldsymbol{u}')$ and $\mathcal{Q}(\boldsymbol{s}, \boldsymbol{u}^*)$. As a result, the update of $Q_1(\tau_1, u_1^*)$ is dominated by non-optimal samples.

## G  NETWORK STRUCTURE OF QFRIS

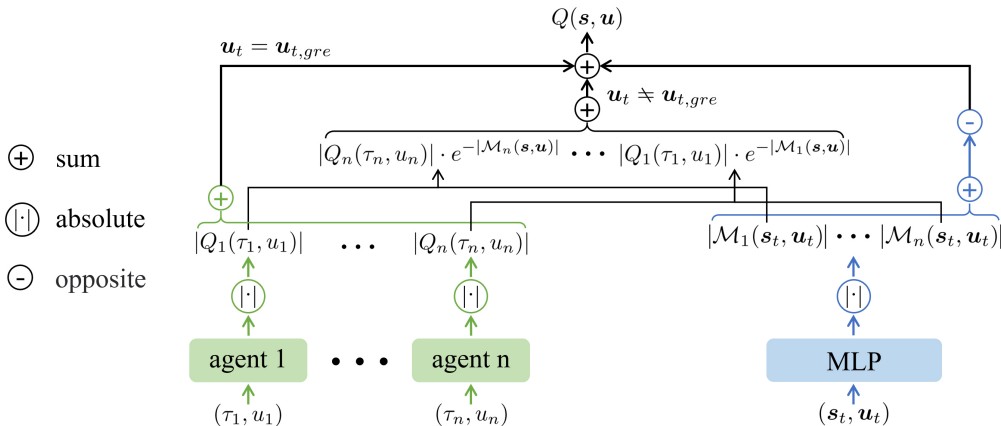

Figure 5: The network structure of QFRIS.

## H  EXPERIMENTS

### H.1  EXPERIMENTAL SETTINGS

In the experiments on one-step matrix games and the verifications of the propositions, all modules are implemented by multilayer perceptrons. A replay buffer of length 1000 is applied for all algorithms. In experiments on predator-prey and SMAC, we adopt the default settings for VDN, QMIX, QPLEX and WQMIX. The length of replay buffer is 5000 and the batch size is 32. For WQMIX, we adopt a weight of 0.5 for predator-prey and 0.1 for SMAC to the samples of poor performance, respectively. The game version of StarCraft II is 69232. Each algorithm is trained for 2e6 steps in MMM2, 2c_vs_64_zg, 3s_vs_5z and 5m_vs_6m, with $\epsilon$ damping from 1 to 0.05 in the first 5e4 steps. Besides, in 6h_vs_8z and 3s5z_vs_3s6z, each algorithm is trained for 5e6 steps, with $\epsilon$ damping from 1 to 0.05 in the first 1e6 steps. All experiments are repeated over 5 seeds.

According to Definition 1, the tasks involving cooperative rewards or interactive transitions of all agents are indecomposable, e.g., predator-prey (Böhmer et al., 2020) and Starcraft multi-agent challenges (SMAC) (Samvelyan et al., 2019). The former involves punitive rewards for miscoordination

of all agents and the latter involves the transition on the health of enemy unites, which is determined by the policies of all agents.

## H.2 MATRIX GAME

In this subsection, we compare the performance of different value factorization functions in one-step matrix games. The pay-off matrix is shown in Fig.H.2(a). Two agents select actions from $\{0, 1, 2\}$ and receive a reward according to the pay-off matrix. To evaluate whether the function is capable to drive the joint Q value function out from the local optimum. We only consider the cases where the greedy action is trapped at $(2, 2)$ after few rounds of training. The experimental results are shown in Fig.H.2. The red curves denote the mean test return. The orange and blue curves denote the difference between the local Q values of the optimal action and current greedy action. The green and brown curves denote the non-linear coefficients on the optimal local Q values contributed by the optimal and non-optimal true Q values, respectively. We do not measure the coefficients on the local Q values for (extended) linear value factorization functions (QMIX and Qtran), where $\frac{\partial \mathcal{F}}{\partial Q_i}$ is a constant. We compare the value factorization functions of QMIX (Rashid et al., 2018), Qtran (Son et al., 2019) (Eq.45), QPLEX (Wang et al., 2020b) (Eq.50), an variant of QPLEX and our method (Eq.12). The variant of QPLEX is

$$\mathcal{F}(Q_{set}(\boldsymbol{u}), \mathcal{M}_{set}(\boldsymbol{u})) = -\sum_{i=1}^{n} (|\mathcal{M}_1(\boldsymbol{u})| + 1) \cdot [Q_i(u_{i,gre}) - Q_i(u_i)] + \sum_{i=1}^{n} Q_i(u_{i,gre}) \quad (51)$$

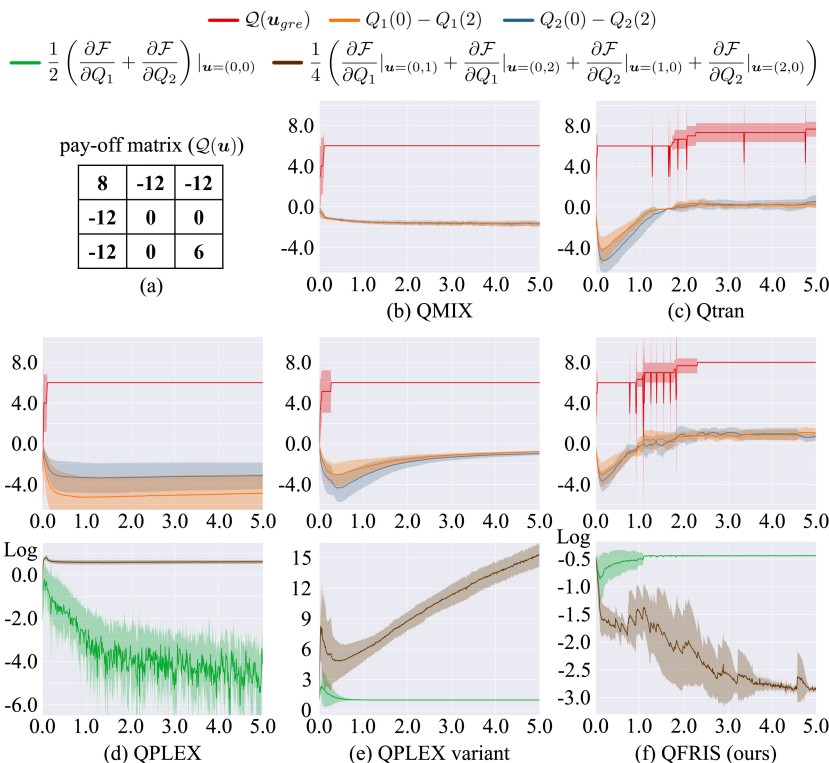

Figure 6: Evaluation of various value factorization operators.

Without complete representation capability (CRC), QMIX suffers from relative overgeneralization. From the red curve of Fig.H.2(a) we can see the greedy action of QMIX becomes $(2, 2)$ at around 0.1k iterations and get trapped in the local optimum. Although Qtran is equipped with CRC, the problem can not be well-solved due to the representation interference on $\mathcal{Q}(\boldsymbol{u}^*)$. The problem of representation interference is more serious in QPLEX. From the green curve of Fig.H.2(d) we can see the non-linear coefficients on $Q_1(0)$ and $Q_2(0)$ contributed by $\mathcal{Q}(0, 0)$ decrease sharply during

training and quickly approximates 0. As a result, $Q_1(0)$ and $Q_2(0)$ grows slowly and the policy is trapped in the local optimum. We add a constant 1 to the non-linear coefficients and obtain a variant of QPLEX (Eq.51). As shown in Fig.H.2(e), the minimum coefficient contributed by $\mathcal{Q}(0,0)$ becomes 1. But the value factorization function still can not drive the joint Q value function out from the local optimum since the coefficients on $Q_1(0)$ and $Q_2(0)$ contributed by non-optimal true Q values (the brown curve) are much larger. QFRIS adopts the representation interference suppression technique, whose coefficients on $Q_1(0)$ and $Q_2(0)$ contributed by non-optimal true Q values (the brown curve) are greatly suppressed, thus is capable to jump out from the local optimum quickly and stably.

## H.3 PREDATOR-PREY

Predator-prey is a cooperative multi-agent task which requires highly coordinated policies. The agents, i.e., the predators are trained to capture the preys moving in random polices. The team is assigned with an instant reward at each time step. The basic reward is 0. There will be a bonus when any prey is captured by more than one agents as well as a punishment if any prey is captured by a single agent. As the punishment increases, the agents are more likely to take a sub-optimal but safe policy, i.e., staying away from the preys. We carry out experiments on 3 different levels of punishments. The experimental results are shown in Fig.3.1.

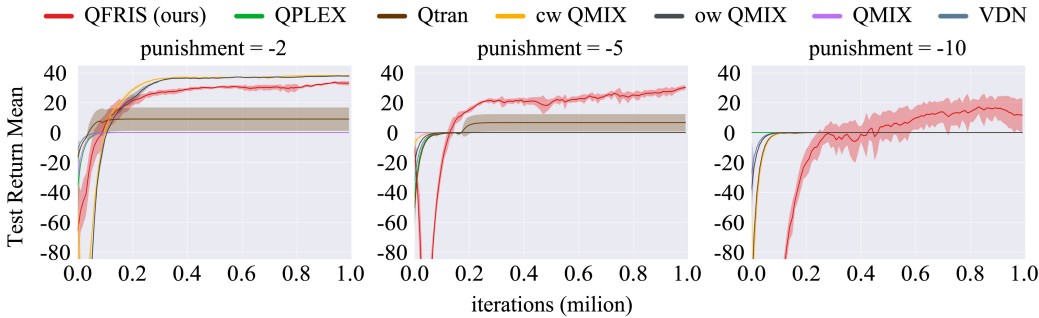

Figure 7: Comparison of value factorization methods on predator-prey with punishment -2, -5 and -10.

From Fig.3.1 we can see that our method can handle the task under different levels of punishments. Two implementations of LVF, i.e., VDN and QMIX are incapable to solve the tasks. cw-QMIX and ow-QMIX reduce the weight of samples with poor performance, which is able to deal with small punishments. Although Qtran and QPLEX adopts value factorization functions that satisfy both IGM and CRC conditions, the problem can not be well solved due to the representation interference.

## H.4 STARCRAFT MULTI-AGENT CHALLENGE

We compare our method with baselines on challenging tasks of StarCraft Multi-Agent Challenge. The experimental results are shown in Fig.H.4. From Fig.H.4 we can see that our method outperforms the baselines in most of the tasks.

## H.5 ABLATION STUDIES

To evaluate the effect of interference suppression introduced by the non-linear value factorization function. We compare QFIRS with a linear variant of it, whose joint Q value function is

$$Q(\boldsymbol{s}, \boldsymbol{u}) = \sum_{i=1}^{n} Q_i(\tau_i, u_i) - |\mathcal{M}(\boldsymbol{s}, \boldsymbol{u})| \cdot I(\boldsymbol{u} = \boldsymbol{u}_{gre}) + V(\boldsymbol{s}) \tag{52}$$

The experimental results are shown in Fig.H.5.

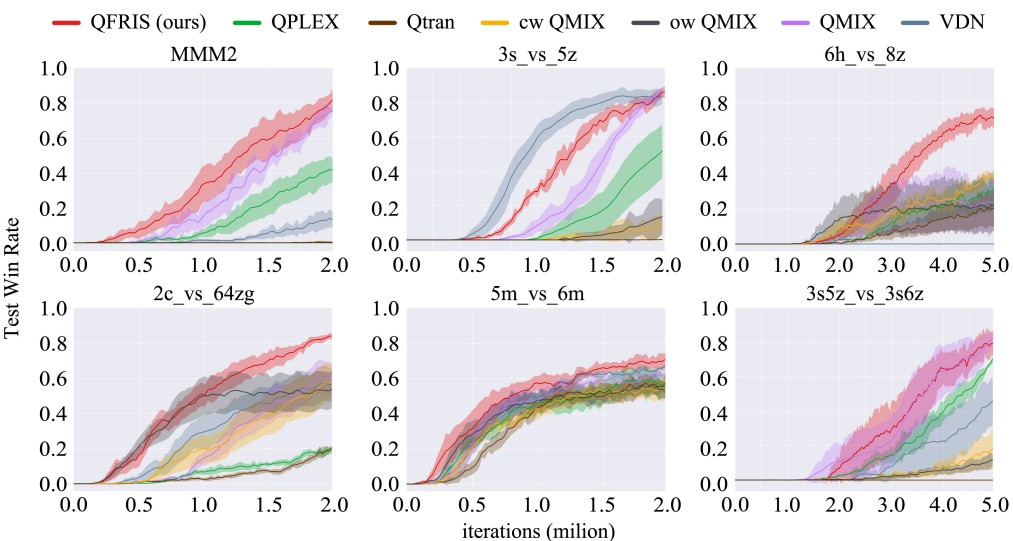

Figure 8: Mean test win rate of value factorization methods on SMAC.

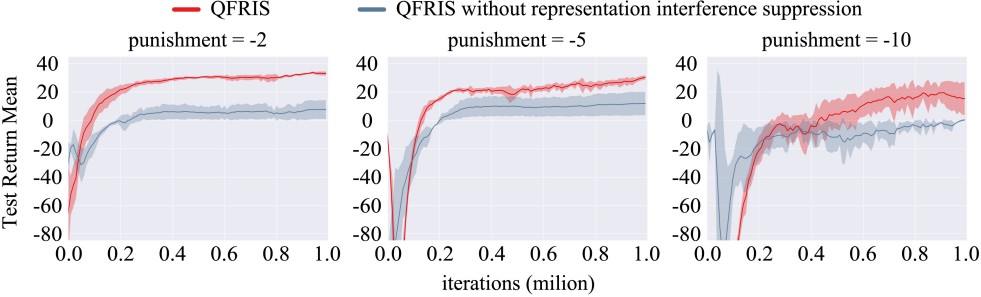

Figure 9: QFRIS vs QFRIS without representation interference suppression on predator-prey.

