# OpenReview forum: "Representation Interference Suppression via Non-linear Value Factorization for Indecomposable Markov Games"
_ICLR.cc/2023/Conference — Submitted to ICLR 2023_

### Official Review · Reviewer_2Pac · 2022-10-20

**Confidence:** 5
**Correctness:** 1
**Technical Novelty And Significance:** 2
**Empirical Novelty And Significance:** 2
**Recommendation:** 1

**Clarity, Quality, Novelty And Reproducibility:**

This paper studies value factorization in Markov games but cannot even provide a mathematically correct definition of decomposable Markov games. For example, in Definition 1, what do you mean by $S_i \times \hat{U}_i$ being a ''subspace'' of the joint state-action space $S_i \times U_i$? According to your later wording $S_i\times \hat{U}_i\subset S\times U$, you seems to mean ''subset'' instead of subspace. However, if you are using ''subset'' and ''subspace'' interchangeably, then the relation in the first bullet point in Definition 1 makes no sense at all.

Getting extremely confused with Definition 1, I referred to the original definition of decomposable MG in Dou et al., 2022 (which is also cited in Definition 1 by this submission) and found it mathematically sound and also much clearer. I do not see how the submission invented this wrong definition but still attributed it to Dou et al., 2022. Also, the place where the submission cited (Dou et al., 2022) is wrong. It should be immediately after Decomposable Markov Game instead of Markov games.

There are tons of typos and vague claims in the submission. Just in Sec 2, I found:

1. $\mathcal{T}_i = (Z_i \times U_i)$:  a trajectory can contain more than one observation-action pair
2. What do you mean the true Q value function? With respect to which policy? The optimal policy or any policy?
3. In Equation (1), you write $Q_i(\tau_i,u_i)$ but below you write $Q_i(u_i,\tau_i)$
4. what do you mean by "The IGM principle enables the coordination of local policies under the centralized trained joint Q value function"
5. what do you mean by "As a result, the optimal Bellman operator could be not a γ−constraint (Wang et al., 2020a) when faced with non-linear true Q value functions. In words, there could be multiple convergences for the joint Q value function (Wan et al., 2021) and the policy would get trapped in sub-optimums." Please explain what is γ−constraint otherwise what's the meaning of adding this sentence here? What do you mean by "multiple convergences"?



**Strength And Weaknesses:**

The writing of this paper is extremely poor and there is a considerable amount of notation/technical errors.



**Summary Of The Paper:**

This paper studies value factorization in decentralized Markov games.


**Summary Of The Review:**

This paper is far below the bar of acceptance and should be immediately rejected. Besides, I strongly urge the authors to carefully revise the draft before the next submission.

---

> ### Author Response · Authors · 2022-11-18
> **Response to Reviewer 2Pac**
>
> Thanks for the comments. We will carefully evaluate the advises. Our paper is not prepared for resubmission. Here we answer the problems and concerns of the reviewer.
>
> > Definition 1:
>
> Please refer to the answer to Reviewer 1.
>
> > Vague claims:
>
> 1. trajectory should contain more than one observation-action pair
>
> Thanks for correction, we have referred the definition in WQMIX, it should be $\mathcal{T}_i \equiv (Z_i \times U_i)^*$
>
> 2. Definition of true Q value function
>
> The true Q value function is the target of value iteration in bellman Equation. Formal definition of the true Q value function can be found in Preliminary.
>
> 3. typos
>
> thanks for correction. We will make a thorough check to revise such problems.
>
> 4. the description of IGM
>
> The IGM principle ensures the optimal consistency between local policies and the joint policy, i.e., $\mathop{\arg\max}\limits_{\textbf{u}}\ Q(\boldsymbol{\tau},\textbf{u}) = \{\mathop{\arg\max}\limits_{u_1}\ Q_1(\tau_1,u_1),\cdots,\mathop{\arg\max}\limits_{u_n}\ Q_n(\tau_n,u_n) \}$
>
> 5. $\gamma$-constraint
>
> We miss the backgrounds of these two citations here.

---

### Official Review · Reviewer_Jjqv · 2022-10-21

**Confidence:** 3
**Correctness:** 3
**Technical Novelty And Significance:** 3
**Empirical Novelty And Significance:** 3
**Recommendation:** 5

**Clarity, Quality, Novelty And Reproducibility:**

## Clarity and Quality
As above, the current version of the paper suffers some clarity and writing issues.
## Novelty
The paper makes nice theoretical observations and empirical contributions, novelty seems fair.
## Reproducibility
Reproducibility is missing and no code is submitted.

**Strength And Weaknesses:**

# Strength:
* The paper makes several nice technical contributions:
1. The paper first identifies the equivalence between linear function factorization and decomposability of the Markov game, thus motivating the study on indecomposable Markov game and non-linear function factorization.
2. The paper identifies the form of extended LVF and argues that one should use the parameterized module form of NVF in order for ensuring IGM and CRC conditions.
3. The paper identifies the representation interference issues in NVF and proposed the new algorithm QFRIS to mitigate this issue.
4. The experiments seem convincing in terms that the results match the theory in the synthetic environment and it achieves competitive performance in the standard benchmarks.
* The paper (main paper + appendix) seems relatively complete, and the technical results seem correct.

# Weakness
* I am having a hard time understanding what is the main message of the paper: section 3 seems a motivation for why we should focus on NVF and section 4 seems to identify how to design NVF that achieves both IGM and CRC. My understanding is that these previous sections serve as motivation for the main algorithm QFRIS, but the methodology section seems pretty short and I am also confused why the benchmark results are deferred to the appendix. Is the empirical success of QFRIS not an important message that the paper tries to deliver? In particular, I am confused about the functionality of several parts of the paper:
1. Section 2.1: why do we introduce POMDP? Do the following sections specifically deal with partial observation?
2. What is the motivation of introducing Lemma 1?
3. And what is the significance of the extended LVF?
* The definition of CRC condition seems vague to me and the proof for some form satisfying CRC seems quite handwavy, but I may miss its connection to some of the formal definitions and arguments.
* The writing of the paper needs some refinement. There are several typos and undefined notations in the current version and some of them requires the readers to guess what the paper really wants to say at some point, for example:
  * In Eq. (1), should $s$ be bold because the above section mentions global variables are bold?
  * typo in the subscript $u^1$ in Eq. (2).
  * Again *u_{t+1}* is not bold under the expectation in Eq. (5).
  * What does it in by optimal trajectory under Eq. (6)? Does it only make sense in deterministic system?
  * The last sentence in the first paragraph in section 4 seems incomplete?
  * Is $u_{gre}$ defined the same way on $\mathcal{M}$ as it is defined on $Q$?
  * What does the subscript in the Q function, for example, $Q_{33}$ in the synthetic environment mean?
  * The figure reference numbers do not seem correct.
* I am not fully convinced by the following argument made under proposition 1: "The joint Q value function of LVF is capable to represent the true Q value function only if each decomposed Markov game of the MGD involves only a single agent." For example, how do you define the reward in the synthetic markov game when it is decomposed to one with only one agent?
* The main text has quite a few discontinuities in the arguments, not just the proofs. For example, it would be better to have at least some overview in section 4.2 on why introducing parametrized functions directly suffers poor convergence, instead of throwing everything into the appendix.

**Summary Of The Paper:**

The paper investigates the relationship between linear value factorization and the decomposability of the Markov game. Then the paper studies the indecomposable Markov game setting and the proposed algorithm, Q Factorization with Representation Interference Suppression, QFRIS, to deal with the representation interference issue in the non-linear value factorization regime. Finally, the paper provides empirical studies on the proposed algorithm and shows that it works well on both the synthetic environment and the benchmark environments.

**Summary Of The Review:**

I am mostly convinced by the technical merits of the paper, but the current version suffers quite a lot of organization and writing flaws. Thus I would recommend a borderline reject at the moment.

---

> ### Author Response · Authors · 2022-11-18
> **Response to Reviewer Jjqv**
>
> Thanks for the reviewer’s comments. The reviewer’s concerns do make sense, which is accepted by the authors. We will carefully evaluate the advises. Our paper is now under a thorough revision, which is not prepared for resubmission. Here we answer the questions and concerns of the reviewer.
>
> > The organize of the paper is unreasonable and main text is discontinuities:
>
> We made a full exploration on the issue of value factorization by answering the questions we continuously confronted with. Our storyline is described as follows:
> In Sec 3.1 we try to find the rule of tasks in which LVF is applicable and inapplicable.
> In Sec 3.2 we explore the property of LVF in tasks which LVF is inapplicable.
> In Sec 4 we explore the general form of value factorization to solve the task which is not solved by VDN and QMIX.
> In Sec 5 we discuss a general problem of the value factorization operators discussed in Sec 4 (e.g., QPLEX and QTran) and propose an operator to solve the problem.
> There are multiple topics in this paper. We are re-organizing the structure of our paper.
>
> > POMDP:
>
> Our experiments are based on partial observable environments. But for the convenience of theoretical analysis, we adopt the full observable assumption. We will add a discussion to fill the gap between partial observation and full observation.
>
> > Lemma 1:
>
> Lemma 1 is a property of decomposable Markov game. We have renamed it to Property 1.
> The decomposition of a Markov game is non-unique. Suppose $(\mathcal{MG}_1, \mathcal{MG}_2,\cdots,\mathcal{MG}_k)$ $(k\geq2)$ is the MGD of Markov game $\mathcal{MG}$. According to Lemma 1, we can obtain all decompositions of $\mathcal{MG}$ by grouping the elements of $(\mathcal{MG}_1, \mathcal{MG}_2,\cdots,\mathcal{MG}_k)$. In fact, the decomposition in Fig.1(c) is derived from the MGD (Fig.1(a)) by Lemma 1.
>
> > Extended LVF:
>
> To discriminate previous mentioned LVF ($\mathcal{F}(\cdot)=\sum_{i=1}^n w_i Q_i(s_i, u_i) + b_s$) from linear value factorization with introduced module M, we name the more general factorization as Extended LVF.
>
> > Typos:
>
> Thanks for correction. We will make a thorough check to revise such problems.
>
> > Reward of the decomposition of Markov game:
>
> In decomposable Markov game, the task is decomposable to multiple sub-tasks.
> Each sub-task (i.e., decomposed game) is also an MDP, which includes a state space, an action space and a reward function. Such reward function is only determined by the action and state of current MDP. For example, Fig.1(e), the reward function of each decomposed game can be described as: when current agent arrives the landmark, reward = 1. Otherwise reward = 0.

---

### Official Review · Reviewer_Nk1r · 2022-10-24

**Confidence:** 2
**Correctness:** 4
**Technical Novelty And Significance:** 3
**Empirical Novelty And Significance:** 3
**Recommendation:** 6

**Clarity, Quality, Novelty And Reproducibility:**

- This paper is well-written and easy to follow.
- Most of the parts is clear, except for the definition of $\hat {\mathcal U}$
- I appreciate the novelty of the decomposable Markov Game, but I'm not sure I understand the intuition about the function $M$ here. It would be more than helpful if the author could make it more clear.
- I did not find the implementation for reproducibility check

**Strength And Weaknesses:**

### Strength:

- The proposed decomposable Markov Game is novel, and using a parameterized function $M$ to fill the gap between the indecomposable and decomposable Markov game is interesting

### Weakness:

- It's not clear to me how we select the subspace components $\hat {\mathcal U}$, despite its existence. If we don't know $\hat {\mathcal U}$, how should we define the parameterized function approximation $M$
- It seems that the $M$ is designed only to fix the gap between the decomposable MG and indecomposable MG? It would be better to highlight the intuition of this $M$.

**Summary Of The Paper:**

This paper studies multi-player Markov games. The authors propose a concept of decomposable Markov game and factorized value function. For the indecomposable Markov games, the authors propose a new method to approximate the global value function by combining the decomposable components and some parameterized module $M$ to fix the gap between the global value function and the combination of decomposable parts.

**Summary Of The Review:**

This paper studies the decomposable Markov game and proposes a new method for approximating the indecomposable one. My major concern is mentioned above. Despite that, I appreciate its novelty and would recommend accept

---

> ### Author Response · Authors · 2022-11-18
> **Response to Reviewer Nk1r**
>
> Thanks for the reviewer’s comments. Our paper is now under a thorough revision, which is not prepared for resubmission. Here we answer the questions and concerns of the reviewer.
>
> > Selection of $\hat{\mathcal{U}}_i$:
>
> the selection of $\hat{\mathcal{U}}_i$ requires priority knowledge of the decomposable way of the game. We need to know how the game is decomposable by agent groups, then we build our approximator with the inputs (i.e., $\hat{\mathcal{U}}_i$) of grouped agents.
>
> > M:
>
> Indecomposable Markov games requires complete representation, which does rely on the introduced module M.

---

### Official Review · Reviewer_cFnT · 2022-10-25

**Confidence:** 4
**Correctness:** 1
**Technical Novelty And Significance:** 2
**Empirical Novelty And Significance:** 2
**Recommendation:** 1

**Clarity, Quality, Novelty And Reproducibility:**

Lots of claims, definitions, and notations in the paper are problematic.

> Definition 1

The reviewer is not convinced that this definition makes sense.

- Dou et al., 2022 gives the definition of decomposable Markov games. Why the authors cite it after Markov game?

- A subspace is better to be defined as a tuple, or the authors need to be clear that they want to define on a subspace of the action-oberservation space. Moreover, what does "a collection of" mean?


>  Linear Value Factorization (LVF) recently gains growing attention, e.g., VDN (Sunehag et al., 2017) and QMIX (Rashid et al., 2018).

QMIX is not a linear value factorization method. Monotonic factorization function class is larger than the linear decomposition function class.

> The IGM principle is defined as the identity between the joint Q value function and the set of factorized local Q value functions

This is also misleading. What does identity mean? IGM only requires the maximum of joint and local Q values are the same..

> Note that we use bold symbols to denote the global and joint variables, e.g., $\mathbf{\mathcal{S}}$ and $\mathbf{\mathcal{u}}$.

Here the authors are defining Dec-POMDP, $\mathbf{\mathcal{S}}$ and $\mathbf{\mathcal{u}}$ here are not variables. They are state and action sets.

> Linear Value Factorization (LVF) ... and becomes the most popular value factorization method in recent years.

The reviewer is surprised by this claim. Linear decomposition is good for theoretical analysis, but, empirically, its performance is not good. The authors should try to follow the frontier of MARL.

> could be multiple convergences for the joint Q value function (Wan et al., 2021) and the policy would get trapped in sub-optimums.

Citation here is problematic.

**Strength And Weaknesses:**

There are many errors in even the basic definitions in this paper. It is very difficult to follow the paper and evaluate its contribution.

**Summary Of The Paper:**

This paper studies a sub-class of Markov games that is linearly decomposable.

**Summary Of The Review:**

The direction may be promising, but please submit a mathematically sound (at least a complete) version of this paper.

---

> ### Author Response · Authors · 2022-11-18
> **Response to Reviewer cFnT**
>
> Thanks for the reviewer’s comments. We are glad to accept the critics and revising our paper accordingly. Our paper is now under a thorough revision, which is not prepared for resubmission. Here we answer the questions and concerns of the reviewer.
>
> > Definition 1:
>
> In the paper we cite, (1) the decomposability of a Markov game is confusing. To proof the linear factorizability of the Q value function, the authors linearly factorize the transition function, which is not linearly factorizable. (2) Decomposability of the Markov game is a sufficient condition of the linear factorizability of Q value function. (3) The decomposition of a game is totally abstract, which does not imply any physical significance. (4) Each game is decomposed into a set of single-agent cases.
>
> In our definition, by contrast, (1) we do not require the unreasonable decomposability of the transition function. (2) We prove that our defined decomposability of Markov game is a **sufficient and necessary condition** for the linearly factorizability of Q value function. (3) The decomposition of a Markov game refers to a set of games, as shown in Fig.1. (4) Each game can be decomposed by agent groups (not only single-agent games), as shown in Fig.1(c).
>
> By our definition, we can discriminate in which tasks VDN is applicable and inapplicable. The definition in the paper we cite is not practical.
>
> > QMIX is Monotonic:
>
> Thanks for correction. Here we only consider the linear case (we should not cite QMIX here). We have explored the general form of monotonic value factorization operators in section 4.
>
> > IGM principle:
>
> Thanks for correction. We have another description of IGM principle in Section 2.2, where “The IGM principle is defined as the identity of the joint greedy action and the set of local greedy actions.”
>
> > Bold symbols & LVF is the most popular:
>
> Inappropriate and loosely descriptions. Thanks for correction.

---

### Decision · Program_Chairs · 2023-01-20

**Decision:**

Reject

**Justification For Why Not Higher Score:**


The variance of the scores for this paper is large, so I read the paper by myself.

The paper in current version is not easy to follow, with misuse of notations and terminologies, making the significance of the paper difficult to be evaluated.

**Justification For Why Not Lower Score:**

N/A

**Metareview: Summary, Strengths And Weaknesses:**


In this paper, the authors investigated the value function representation in decomposable and indecomposable Markov games. The authors also proposed a novel value function parametrization way and justified the new parametrization on several benchmarks.


The major issues raised by the reviewers lies in several aspects:

- The correctness of the notations and definitions. In fact, as several reviewers pointed out, these errors make the paper difficult to be evaluated.

- The clarity of the presentation. The current presentation does not provide enough emphasis on the major contribution, and thus hide the main message and make the reader confusing.

The paper may contain significant contribution. However, the current presentation makes the paper difficult to follow and the contribution unclear. I suggest the authors to carefully consider the feedback from the reviewers to improve the paper before next submission.